# Elderberry Extracts: Characterization of the Polyphenolic Chemical Composition, Quality Consistency, Safety, Adulteration, and Attenuation of Oxidative Stress- and Inflammation-Induced Health Disorders

**DOI:** 10.3390/molecules28073148

**Published:** 2023-03-31

**Authors:** Ahmed G. Osman, Bharathi Avula, Kumar Katragunta, Zulfiqar Ali, Amar G. Chittiboyina, Ikhlas A. Khan

**Affiliations:** 1National Center for Natural Products Research, University of Mississippi, University, MS 38677, USA; 2Division of Pharmacognosy, Department of BioMolecular Sciences, School of Pharmacy, University of Mississippi, University, MS 38677, USA

**Keywords:** *Sambucus nigra*, *Sambucus canadensis*, extracts, elderberry fruit, chemical composition, analytical techniques, health effects

## Abstract

Elderberry is highly reputed for its health-improving effects. Multiple pieces of evidence indicate that the consumption of berries is linked to enhancing human health and preventing or delaying the onset of chronic medical conditions. Compared with other fruit, elderberry is a very rich source of anthocyanins (approximately 80% of the polyphenol content). These polyphenols are the principals that essentially contribute to the high antioxidant and anti-inflammatory capacities and the health benefits of elderberry fruit extract. These health effects include attenuation of cardiovascular, neurodegenerative, and inflammatory disorders, as well as anti-diabetic, anticancer, antiviral, and immuno-stimulatory effects. Sales of elderberry supplements skyrocketed to $320 million over the year 2020, according to an American Botanical Council (ABC) report, which is attributable to the purported immune-enhancing effects of elderberry. In the current review, the chemical composition of the polyphenolic content of the European elderberry (*Sambucus nigra*) and the American elderberry (*Sambucus canadensis*), as well as the analytical techniques employed to analyze, characterize, and ascertain the chemical consistency will be addressed. Further, the factors that influence the consistency of the polyphenolic chemical composition, and hence, the consistency of the health benefits of elderberry extracts will be presented. Additionally, adulteration and safety as factors contributing to consistency will be covered. The role of elderberry in enhancing human health alone with the pharmacological basis, the cellular pathways, and the molecular mechanisms underlying the observed health benefits of elderberry fruit extracts will be also reviewed.

## 1. Introduction

Elderberry varieties include 20 species belonging to the genus *Sambucus* in the family Caprifoliaceae. Among these species, *Sambucus nigra* and *Sambucus canadensis* are currently used as food colorants owing to their safety and high content of anthocyanins, which are a subclass of polyphenols known for their powerful antioxidant properties and health benefits. The naturally occurring European elderberry (*S. nigra*) is distributed throughout Europe and has been introduced into various parts of the world as a source of elderberry fruit and flowers.

Polyphenol is a plant secondary metabolite, derived from the shikimate-derived phenylpropanoid and/or the polyketide pathway(s), having more than one phenolic ring, and devoid of any nitrogen-containing functionality [1]. The most popularly consumed elderberry species for supporting immunity is black elderberry (*S. nigra*; also known as European elderberry or black elder). Elderberry has long been used as a remedy in traditional medicine in many indigenous cultures [2]. The remedial utilities of elderberry extract include diabetes management, blood pressure regulation, obesity control, memory improvement [3], and immune system enhancement, in addition to alleviation of flu and common cold symptoms. These effects were verified in several in vitro, in vivo studies and clinical trials [4,5,6,7,8]. The medicinal properties of elderberries are reportedly attributable mostly to their high content of polyphenols, including anthocyanins, flavonols, and proanthocyanidins [9]. Elderberry (*S. nigra*) has traditionally been prescribed to prevent or treat respiratory disorders, including the common cold, influenza, and infections due to novel coronaviruses (including severe acute respiratory syndrome (SARS), Middle East respiratory syndrome (MERS), and the coronavirus disease of 2019 (COVID-19) [10].

In recent years, elderberry fruit has received great attention due to the presence of large amounts of anthocyanins, as the predominant constituents, besides other polyphenols, which exhibit antioxidant activity and are known for their health benefits. Research has demonstrated the anti-inflammatory, antiviral, antiproliferative, anti-diabetic, and immunostimulatory effects of elderberries [11]. Elderberries have exceptionally high anthocyanin content compared with other fruit species [602.9–1265.3 mg CGE 100 g^−1^ FW] [12]. Considering the effects of the environmental, growing, and developmental factors, as well as the influence of the different methodologies used in processing, the profiles of the chemical composition of the different cultivars of elderberry are variable at different levels, ranging from narrowly variable to widely variable. Consequently, the analysis of different extracts of elderberry fruit (*S. nigra*) showed that the distribution of the three types of phenolic constituents in elderberry fruit, namely, cinnamic acids, flavonols, and anthocyanins, represents 5%, 13%, and 80%, respectively [13].

The demand for botanical sources rich in antioxidant constituents has been very active for a long time, and current demand is projected to grow at an annual rate of approximately 6% globally during the period 2019–2029 [14]. Elderberry is anticipated to gain considerable attention in the coming years due to its high level of anthocyanins, which have a powerful antioxidant capacity, and the dark-violet elderberry fruit. Being water-soluble natural colorants, anthocyanins have long been used as food colorants, and are generally recognized as safer than synthetic colorants [15]. According to the U.S. Food and Drug Administration (FDA) Code of Federal Regulations (CFR), fruit (21CFR73.250) and vegetable (21CFR73.260) juice concentrates are approved for use as natural food colorants. Nevertheless, the utility of anthocyanins as natural colorants is restricted by their low stability and color fading. In contrast, the American elderberry (*S. canadensis*) contains large amounts of acetylated anthocyanins, which have augmented stability against changes in Ph and exposure to heat and light [13]. Moreover, the majority of elderberry extracts are derived from fruit, although in a few cases the extracts are derived from flowers. These elderflowers are not prevalently used as botanical supplements for enhancing the immune system as the flowers of most cultivars were found to be devoid or have only traces of anthocyanins, which are the desired polyphenols for enhancing health, particularly against oxidative stress- and age-related disorders [16,17,18].

In recent years, elderberry fruit has received great attention due to the presence of large amounts of anthocyanins and other polyphenols. Several investigations have demonstrated the anti-inflammatory, antiviral, antiproliferative, anti-diabetic, and immunostimulatory effects of elderberries [11]. The American elderberry (*S. canadensis*) is a plant native to North America and bears anthocyanin-rich fruit. It grows wild throughout the eastern United States as well as the Midwest, and can be seen along roadsides and rivers. The American elderberry is rarely consumed as fresh fruit and is used mostly in processed food and beverages. This might be attributed to its rather small fruit size and tart taste due to its high content of polyphenols [9]. The high abundance of polyphenols in the European elderberry (*S. nigra*) and particularly the anthocyanins, which represent the major constituents of elderberry and are reportedly known for their powerful antioxidant and anti-inflammatory activities [19] justify the use of elderberry in the treatment of several health disorders. These include viral infections and upper respiratory infections, colds, flu, and catarrh, as well as use as a diuretic or to alleviate pain [8,20,21,22]. Elderberry extract has also demonstrated beneficial effects related to cardiovascular diseases (CVDs), cancer [23], and diabetes, as well as having anti-inflammatory properties [24] and immune-stimulating, anti-depressive, chemo-preventive, and atheroprotective effects [13,20,25]. In addition to the abovementioned subjects, the following aspects will be addressed in the current review, namely, the health benefits and the remedial uses of the more common European elderberry (*S. nigra*) and the less popular American elderberry (*S. canadensis*).

This review also examines the health benefits of elderberry pertaining to chronic medical conditions, as opposed to viral respiratory infections, which are seasonal, given that the effectiveness of elderberry is not substantiated by unequivocal clinical evidence [10].

The overall health benefits of elderberry are largely centered on treating, reducing the severity, or delaying the onset of oxidative stress- and inflammation-mediated chronic health disorders. This is because the underlying cellular and molecular mechanisms of several pharmacological effects of elderberry, other berries, and their major constituents (anthocyanins) are based on their antioxidative and anti-inflammatory properties. The cellular and molecular mechanisms that mediate the pharmacological effects of elderberry or elderberry anthocyanins will be discussed.

Analytical methods and techniques include infrared spectroscopy (IR), nuclear magnetic resonance (NMR), high-performance thin layer chromatography (HPTLC), capillary electrophoresis (CE), gas chromatography (GC), and liquid chromatography/mass spectrometry (LC-MS). These methods are reviewed to provide appropriate tools for standardizing the extracts and ascertaining the quality consistency and the chemical safety of botanical products containing elderberry, either as single-ingredient products or as an ingredient of multi-ingredient products. It is worth noting that gas chromatography-mass spectrometric analysis is employed to identify monoterpenes, sesquiterpenes, and other lipophilic constituents of elderberry as marker compounds to discriminate between the different cultivars and study the factors that affect the chemical composition of elderberry.

A literature search was conducted using electronic databases from SciFinder, Google Scholar, and PubMed. The search covered the period 2000–2022. The following search terms were used: elderberry, *Sambucus nigra*, *Sambucus canadensis*, *Sambuci fructus*, European elder, European elderberry, black elderberry, European black elderberry, and American elderberry.

The search conducted using the three search engines resulted in the following: SciFinder led to 785 results for *S. nigra* and 43 results for *S. canadensis*; PubMed led to 975 results for *S. nigra* and 17 results for *S. canadensis*; and Google Scholar led to 24,300 results for *S. nigra* and 13,500 results for *S. canadensis*. Of these results, 80 were selected based on their relevance to the different sections of the review.

The full-text articles were evaluated and screened based on inclusion and exclusion criteria. Inclusion criteria were: articles published in the English language; articles reporting on the fruit/berry as the target plant part since the majority of the elderberry extracts utilized for manufacturing elderberry dietary supplements are made from the berries; articles reporting on the two main species for manufacturing extracts, namely, *Sabmucus nigra* (European elderberry) and *Sambucus canadensis* (American elderberry); articles reporting only on the polyphenolic constituents of the two main species; and articles reporting on the pharmacological effects aiming to attenuate oxidative stress and suppress inflammatory conditions.

Exclusion criteria were: species other than *Sambucus nigra* and *Sambucus canadensis*; plant part (plant organ) other than berries (fruit); chemical constituents other than polyphenols; and aqueous extracts that contain proteins or polysaccharides. Exclusion criteria for clinical trials were: trials on mixed berries; trials combining elderberry with other ingredients, such as Zn, vitamin C, or Echinacea extracts; and trials other than those related to the effect of elderberry extracts on cardiovascular disease, hyperlipidemia, or diabetes.

## 2. Factors Affecting the Consistency of the Chemical Composition of the Herbal Extracts

The chemical composition of elderberry extracts is influenced by intrinsic factors, including genetic elements that characterize the cultivars, the season of flowering, the season of fruit ripening, and the degree of ripeness. The chemical composition is also affected by extrinsic factors that may be divided into preharvest and postharvest factors. The preharvest factors are divided into growing conditions and environmental factors [26]. Growing conditions include edaphic factors [27,28,29] and harvesting activities [30,31,32]. The environmental factors include biotic and abiotic stress [33], altitude [34,35], exposure to light, temperature, and rainfall. Among preharvest factors, it has been found that the elderberry fruit has higher polyphenolic content when grown in a well-organized orchard than in the wild. Postharvest factors include the drying method of the plant raw material, storage conditions, and extraction methods. In addition, there are external factors that include substitution with unintended species, adulteration, and contamination. Processing, especially thermal treatment, could result in the degradation of the elderberry bioactive constituents such as anthocyanins [36].

Thus, it has been demonstrated that the chemical composition of an elderberry fruit cultivar is affected by combinations of intrinsic and extrinsic factors, which result in variations in its chemical profile from one year to another [22]. It has been established that each particular cultivar is associated with a unique chemical profile [13,37,38]. Quality consistency is essential for safety assurance and for consistency of the medicinal effects of herbal products. Quality consistency can be ascertained by using advanced analytical tools. The utility of these analytical tools will be discussed in the following sections. Poor quality is due overwhelmingly to a lack of appropriate quality assurance and quality control practices, beginning with the plant raw material and continuing through to production of the extracts. Quality refers to the established specifications for identity, purity, chemical composition, and limits on contaminants, all to ensure consistency in the quality of herbal products. Purity specifications are established to protect against the possibility of contamination, adulteration, and the presence of harmful substances [39]. The quality assurance and the quality control of the plant raw materials contribute significantly to improving the quality consistency of the end products.

### 2.1. Cultivar and Harvesting Year

In a comparative analysis conducted in Hungary on the chemical composition of the fruit of 11 elderberry cultivars, the total polyphenol content (TP) and total anthocyanin content (TA) was determined over three consecutive years. Significant differences in TP and TA were found between the cultivars, with several cultivars showing high content of TP and TA. The TP of the 11 elderberry cultivars harvested over three years revealed vast variations. In addition, the data were inconsistent with previously reported data, where some studies reported lower values [37], while others reported higher values [13]. However, the current results, with their high content of TP, were consistent with a previous study [40]. The variability in the TP could be attributed to the genotype, the harvesting year, or other environmental conditions in determining the chemical composition. Further, statistical analysis indicated that the high content of TP can largely be ascribed to the significant effect of the genotype.

The total anthocyanin content also showed substantial variability among the 11 cultivars. The data were again inconsistent with the literature, reporting either higher [41] or lower values [37]. Statistical data also indicated the significant effect of the cultivar on mean anthocyanin content. These values are summarized in Table 1 [12].

### 2.2. Effect of the Harvesting Year

In addition to the discussion above, a recent investigation reported that the harvesting year had a stronger effect on the chemical composition than the cultivar, including total sugars, anthocyanins, and phenolic compounds [13]. The investigation was conducted on the main Portuguese cultivars, “Sabugueiro”, “Sabugueira”, and “Bastardeira”, over three consecutive years. The cultivar “Bastardeira” contained the highest level of anthocyanins and polyphenols, and demonstrated the highest antioxidant activity compared with the other cultivars. Significant differences in the total phenolic content (TPC), total anthocyanin content (TAC), and the *ortho*-diphenol content (ODPC) were observed in the three cultivars. The TPC of the three Portuguese cultivars was found to be higher than in previously reported data [37]. Similarly, the TAC, of the three Portuguese cultivars showed higher values than other reported values, including those recorded for harvesting elderberry fruit in the years 2004 and 2005 [37], and data on elderberry fruit obtained from Turkey in 2009 [20]. The effect of the harvesting year on the yields of TPC, TAC, and ODPC in “Bastardeira”, the cultivar containing the highest content of anthocyanins and polyphenols, is presented in Table 2.

The HPLC analysis of the phenolic content of the elderberry fruit cultivars revealed identical profiles at the qualitative level between the different cultivars and over the harvesting years; however, there were differences at the quantitative level. In this investigation, ten phenolic compounds were identified. These compounds were (1) cryptochlorogenic acid, (2) chlorogenic acid, (3) quercetin 3-*O*-glucoside, (4) quercetin 3-*O*-rutinoside, (5) quercetin, (6) isorhamnetin 3-*O*-glucoside, (7) cyanidin 3,5-*O*-diglucoside, (8) cyanidin 3-*O*-sambubioside 5-*O*-glucoside, (9) cyanidin 3-*O*-glucoside, and (10) cyanidin 3-*O*-sambubioside. Neochlorogenic acid was identified in extracts of the 3rd-year harvested cultivars. The study also showed the significant effect of the harvesting year on variations in the level of the different phenolic compounds, with the exception of cyanidin 3-*O*-glucoside, whose levels in elderberry fruit did not change over the three harvesting years.

This study reported chlorogenic acid in elderberries as the major cinnamic acid in the third harvesting year only, and cryptochlorogenic acid as the major cinnamic acid derivative in the first and second years. The variability in the chemical composition of the elderberry fruit cultivars over the three years may be attributable to changes in environmental conditions, such as rainfall, temperature, and exposure to sunlight as the plants are highly responsive to environmental conditions.

### 2.3. Influence of Cultivar on Elderberry Chemical Composition

Despite the significant influence of the harvesting year on the chemical composition of the elderberry fruit, it was observed that certain constituents of the cultivars, including cyanidin 3-*O*-sambubioside-5-*O*-glucoside, fructose, malic acid, cyanidin 3,5-*O*-diglucoside, glucose, and fumaric acid, are characteristic of specific cultivars. Thus, they can be used as markers to distinguish between the different cultivars [13].

### 2.4. Effect of Ripening Stage

A recently published study investigated the effects of cultivar and stage of ripeness on the accumulation of anthocyanins and other constituents in American elderberry. Five cultivars were examined for their anthocyanins and other phenolic content during the different stages of ripening. It was found that the phenolic content significantly increased during fruit ripening. The monomeric anthocyanin content was reduced in half-ripe berries. However, these monomeric anthocyanins increased in ripened fruit. The rate of increase of the anthocyanin content was higher than that of the phenolic content. However, the polymeric anthocyanin content was the only phytochemical that decreased during ripening. Upon ripening, the anthocyanin content was found to account for 80% of the total phenolic content of the fruit. The anthocyanin profile of fully ripe fruit was identical to that of half-ripe fruit, except for the absence of cyanidin 3-(*trans*)-*O*-coumaroyl-glucoside. The *cis* isomers of cyanidin 3-*O*-coumaroylsambubioside 5-*O*-glucoside were identified in half-ripe fruit as well as in fully ripe fruit. Additionally, acylated anthocyanins constituted approximately 80% of the total polyphenols in the half-ripe fruit and approximately 70% in the fully ripe fruit. However, both acylated and non-acylated anthocyanins were much more abundant in the fully ripe berries than in fruit at other ripening stages. Comparing the change in the anthocyanin content with that in the phenolic content, it was observed that the anthocyanin/phenolic content ratio increased from approximately 20% at the half-ripening stage to 85% at the fully ripening stage. However, while the total anthocyanin content increased, the polymeric anthocyanin content decreased. In contrast, the total phenolic content did not increase during ripening, while the acylated anthocyanin content increased at the half-ripening stage [9].

### 2.5. Effect of Source Acquisition: Cultivated Versus Wild-Collected Elderberry

In a study on black elderberry comparing the chemical composition of fruit harvested from cultivated plants with the chemical composition of fruit derived from wild plants, four cultivated and six wild plants were analyzed for their anthocyanin content and profiles using HPTLC-ESI-MS. In this investigation, it was concluded that the anthocyanin content was higher in berries of cultivated fruit than in wild-collected fruit. Additionally, in both fresh and dried fruit from wild-collected plants, the content of the major anthocyanins, cyanidin 3-*O*-sambubioside, and cyanidin 3-*O*-glucoside was much lower than in the cultivars [21].

### 2.6. Effect of Geographic Region

In a study on the American elderberry (*S. canadensis*) to identify variations in the content of selected phenolics, fruit from 107 wild elderberry plants from 18 states in the USA was included in the study. The levels of three major flavonols of elderberry fruit, namely, rutin, quercetin, and isoquercetin, as well as chlorogenic acid, were determined using HPLC-UV. The mean total content of the three flavonols and chlorogenic acid was 71.9 mg/100 g FW, with a variation of 7.0 to 209.7 mg/100 g FW within each collection. It was observed that elderberry fruit collected from the southeastern regions had significantly higher content compared with fruit collected from the northern regions. Considerable variation in the levels of the individual flavonols and chlorogenic acid profiles of the berries were observed in the wild-collected fruit, with ranges of 3.5–170 mg/100 g, 0.7–48.5 mg/100 g, 0.6–45.4 mg/100 g, and 0–25.7 mg/100 g, respectively. Additionally, the flavonol content of the individual collections was highly fluctuating, with the highest value being 209.7 mg/100 g FW and the lowest being 7.0 mg/100 g FW––approximately a 30-fold difference in flavonol content in the wild-collected elderberry fruit samples. The enormous variation in the chemical composition of these wild-collected samples can be ascribed to the influence of a combination of factors, including genetic and environmental factors [42].

### 2.7. Effect of Altitude

The effect of altitude on the phenolic chemical composition of black elderberry fruit (*S. nigra*) was investigated using wild-collected samples of elderberry fruit grown at two different altitudes (670 m and 1000 m above sea level). Using RP-HPLC-PDA, three samples from each altitude were analyzed for phenolic content, including flavonoids, anthocyanins, and hydroxycinnamic acid derivatives. The studied samples were collected over two consecutive years. The results of this study showed that the level of the anthocyanin, cyanidin 3-*O*-glucoside, decreased as the elevation increased (0.86 mg/100 mg DW (dry weight) at 670 m to 0.69 mg/100 mg DW at 1000 m) (year 1), and (0.95 mg/100 mg DW at 670 m to 0.68 mg/100 mg DW at 1000 m) (year 2). The concentration of the flavonol, rutin, exhibited the opposite response to elevation, with its content increasing as the altitude increased (0.33 mg/100 mg DW at 670 m to 0.39 mg/100 mg DW at 1000 m (year 1) and (0.25 mg/100 mg DW at 670 m to 0.34 mg/100 mg DW at 1000 m (year 2) [43].

### 2.8. Adulteration

The surge in popularity of elderberry extract, in combination with a shortage in the supply and an increase in the cost of the raw herbal material, as well as the cost of manufacturing the extract, led to the emergence of adulteration. Consequently, low-quality or adulterated elderberry has appeared on the market. To achieve the highest possible level of certainty in the results of analyzing herbal samples to detect adulteration, using multiple techniques and several quality markers is the correct approach to follow. Moreover, the unambiguous detection of adulteration and the identification of adulterants is highly reliant on pursuing the approach of coupling untargeted and targeted metabolomics. This approach produces comprehensive chemical profiles, making it possible to discriminate the adulterants from the genuine samples [16].

The prevalent adulterant has been identified as black rice extract, along with other unidentified materials used as adulterants. A review of the literature, in addition to data collected from elderberry suppliers, manufacturers, and contract analytical laboratories, provided evidence that the adulteration of elderberry extracts is quite common. In a recent study using UHPLC-PDA-MS to analyze 31 dietary supplements [16], it was found that more than 60% of the dietary supplements claiming to contain European elderberry displayed significantly different anthocyanin profiles from the authentic elderberry anthocyanin profile, indicating adulteration with black rice, purple carrot, and *S. nigra* flowers. Nevertheless, several reports suggest that the elderberry extracts themselves have been utilized as adulterants [44]. Recently, it has been observed that adulterators are using sophisticated approaches to deceive the analytical methods in order to evade the detection of adulterants. Consequently, assessing the authenticity of *S. nigra* which is frequently adulterated with black rice extract by analyzing its anthocyanin profile using methods such as UV/vis, HPTLC, and HPLC-UV/vis will put these adulteration methods at risk of detection [45].

## 3. Polyphenolic Chemical Composition of Black Elderberry Fruit

Reviewing the literature revealed that the chemical composition of the black elderberry fruit (*S. nigra*) shows variation in the different reports at the qualitative and/or the quantitative levels. The variability of the chemical profiles of elderberries is attributed to the influence of several factors, including genetic, climatic, soil conditions, cultivar, harvesting year, growing conditions, and stage of ripening. Consequently, the current review aims to present the widely reported individual polyphenols, which constitute the major and the most bioactive components that contribute to the health-promoting effects of black elderberry and the American elderberry. Cyanidin derivatives are reported as the major anthocyanins present in the fruit of *S. nigra.* As shown in Figure 1, the predominant cyanidin-based anthocyanins are cyanidin 3-*O*-glucoside (**1**), cyanidin 3-*O*-sambubioside (**2**), and cyanidin 3-*O*-sambubioside-5-*O*-glucoside (**3**) [16,21]. Traces of pelargonidin, petunidin, and delphinidin derivatives were encountered in some *S. nigra* cultivars [37,40].

A recent study analyzed the polyphenols in 27 samples of *S. nigra* fruit and fruit extracts/fruit juices. In this study, the predominant anthocyanins were found to be cyanidin 3-*O*-glucoside, cyanidin 3-*O*-sambubioside, and cyanidin 3-*O*-sambubioside-5-*O*-glucoside. The relative concentrations of these anthocyanins were variable among the different collections. The caffeoylquinic acids (neochlorogenic acid and chlorogenic acid) and the flavonols (rutin, hyperoside, isoquercitrin, isorhamnetin 3-*O*-rutinoside, and quercetin) were identified in all *S. nigra* berry samples [16]. In a different study, chlorogenic acid, quercetin 3-*O*-rutinoside, quercetin 3-*O*-glucoside, and quercetin 3-*O*-arabinoside were detected in *S. nigra* [46]. Some authors reported the detection of neochlorogenic acid in different geographical regions [37,38]. Chlorogenic acid was identified as the major cinnamic acid derivative in elderberries. Other published studies reported the occurrence of neochlorogenic acid and cryptochlorogenic acid in some cultivars [13,37,38]. Trace levels of (+)-catechin, (−)-epicatechin, flavonols, and phenolic acids were reportedly detected in *S. nigra* berries [13]. Kaempferol was also detected in a very low concentration (0.002/g) in *S. nigra* fruit, along with the phenolic acids, caffeic (0.016 mg/g), *p*-coumaric (0.011 mg/g), and ferulic [47].

While there are three predominant anthocyanins in black elderberry (*S. nigra*), a study investigating the analysis of polyphenols in the fruit of two cultivars and three selections of *S. nigra* L (“Haschberg,” “Rubini,” “Selection 13,” “Selection 14,” and “Selection 25”) using HPLC/MS, led to the identification of five major anthocyanins. These were cyanidin 3-*O*-sambubioside-5-*O*-glucoside and cyanidin 3,5-*O*-diglucoside (**4**) (Figure 1), cyanidin 3-*O*-sambubioside, cyanidin 3-*O*-glucoside, and cyanidin 3-*O*-rutinoside (**5**), with cyanidin 3-*O*-sambubioside as the most abundant anthocyanin, representing more than half of the total anthocyanin content. Additionally, the flavonols, quercetin, quercetin 3-*O*-rutinoside, and quercetin 3-*O*-glucoside, were identified. However, the concentrations of the individual anthocyanins and flavonols varied between the cultivars and the selections [41]. The *S. nigra* cultivar, “Kors0r,” was analyzed for the identification of its individual anthocyanins. Four anthocyanins, cyanidin 3-*O*-sambubioside-5-*O*-glucoside, cyanidin 3,5-*O*-diglucoside, cyanidin 3-*O*-sambubioside, and cyanidin 3-*O*-glucoside, as well as trace levels of pelargonidin 3-*O*-glucoside, were identified. In the same study, the cultivar, “Haschberg,” was found to contain trace levels of cyanidin 3-*O*-rutinoside and delphinidin 3-*O*-rutinoside in addition to the aforementioned anthocyanins identified in the cultivar, “Kors0r” [37]. The polyphenols in the fruit of thirteen elderberry cultivars of black elderberry (S. *nigra*) were analyzed, revealing different chemical profiles composed of chlorogenic acid, neochlorogenic acid, cryptochlorogenic acid, quercetin, quercetin 3-*O*-rutinoside (rutin), quercetin 3-*O*-glucoside (isoquercitrin), kaempferol 3-*O*-rutinoside, kaempferol 3-glucoside (astragaline), isorhamnetin 3-*O*-rutinoside, and isorhamnetin 3-*O*-glucoside, with rutin as the major flavonol. In this study, it was observed that the total anthocyanin content varied nearly threefold among the different cultivars [48]. However, in a geographic cultivar of *S. nigra*, namely, the “Andean” elderberry, cyanidin 3-*O*-glucoside and cyanidin 3-*O*-sambubioside occur as the major anthocyanins, as well as their respective isomers [49]. Furthermore, cyanidin-3-*O*-sambubioside and cyanidin 3-*O*-glucoside were identified in a different report as the major anthocyanins [13].

## 4. Polyphenolic Chemical Composition of American Elderberry Fruit

In a recent study, analysis of the anthocyanin content of *S. canadensis* (American elderberry) resulted in the identification of four acylated anthocyanins, namely, cyanidin 3-*O*-sambubioside-5-*O*-glucoside, cyanidin 3-*O*-sambubioside, cyanidin 3-*O*-glucoside, and cyanidin 3-*O*-[6-*O*-(*E*-*p*-coumaroyl-2-*O*-β-d-xylopyranosyl)-β-d-glucopyranoside]-5-*O*-β-d-glucopyranoside (**6**) (Figure 1), as the major anthocyanins. The relative concentrations of these anthocyanins varied among different collections. The caffeoylquinic acids, neochlorogenic acid and chlorogenic acid, and the flavonols, rutin, hyperoside, isoquercitrin, isorhamnetin 3-*O*-rutinoside, and quercetin, were identified in all *S. canadensis* berry samples [16]. Neochlorogenic acid, chlorogenic acid, rutin, and isorhamnetin 3-*O*-rutinoside were also reported in a previous work on *S. canadensis* [37]. However, in an investigation of the anthocyanin profiles of four elderberry species and interspecific hybrids, the authors reported the existence of three acylated anthocyanins, including cyanidin 3-*O*-(*Z*)-*p*-coumaroyl-sambubioside-5-*O*-glucoside, cyanidin 3-*O*-(*E*)-*p*-coumaroyl-sambubioside-5-*O*-glucoside, and cyanidin 3-*O*-β-d-(*p*-coumaroyl)-sambubioside [9].

A different publication mentioned the presence of kaempferol but did not include isorhamnetin 3-*O*-rutinoside among the flavonols of the American elderberry (*S. canadensis*) [37,43]. In the cultivars, “Adams 2,” “Johns,” “Scotia,” “York,” and “Netzer” of *S*. *canadensis*, trace levels of petunidin 3-*O*-rutinoside were identified [37]. Delphinidin 3-*O*-rutinoside and petunidin 3-*O*-rutinoside have been identified in *S. canadensis,* but not as major anthocyanins [37]. Zhou et al. reported the occurrence of two nonacylated anthocyanins (cyanidin 3,5-*O*-diglucoside and cyanidin 3-*O*-sambubioside-5-O-glucoside) and three acylated anthocyanins (cyanidin 3-*O*-(*cis*)-coumaroylsambubioside-5-*O*-glucoside, cyanidin 3-O-(*trans*)-coumaroyl-*O*-glucoside, and cyanidin 3-*O*-(*trans*)-coumaroylsambubioside-5-*O*-glucoside). All identified anthocyanins were derivatives of cyanidin. Cyanidin 3-*O*-(*trans*)-coumaroylsambubioside-5-*O*-glucoside was the major one, constituting 65–70% of the total anthocyanin content. *p*-Coumaric acid was the only phenolic acid involved in the acylation of anthocyanins found in American elderberry fruit. This co-occurrence of *cis* and *trans* isomeric anthocyanins is rare in edible plant sources [9]. Acylated anthocyanins are a subclass of more stable anthocyanins and occur more frequently in vegetable sources, such as red radish, black carrot, and red cabbage. Berries such as cranberry, blackberry, blueberry, or European elderberry prevalently contain only non-acylated anthocyanins. In contrast, American elderberry, which is available in North America, contains highly acylated anthocyanins, making it an excellent source of natural colorants with several desirable characteristics. Acylated anthocyanins represent approximately 80% of the total anthocyanin content of an S. *canadensis* berry at the half-ripened stage, and approximately 70% when fully ripened [9]. Cinnamic acid-acylated anthocyanins were reported to exhibit enhanced antioxidant activity [50]. Acylated anthocyanins in black carrot exhibited higher storage stability than nonacetylated anthocyanins at three storage temperatures (4, 25, and 40 °C) over a period of 90 days [37].

## 5. Stability of Acylated Versus Non-Acylated Anthocyanins

The content of the acylated anthocyanins in the *S. canadensis* berry represents approximately 80% of its anthocyanin content at the half-ripened stage, and approximately 70% when fully ripened [9]. Cyanidin 3-*O*-(6-*O*-*E*-*p*-coumaroyl-2-*O*-*β*-d-xylopyranosyl)-*β*-d-glucopyranoside-5-*O*-*β*-d-glucopyranoside in *S*. *canadensis* was found to be more stable than the non-acylated cyanidin 3-*O*-sambubioside in *S*. *nigra*. The acylation of anthocyanins was reportedly responsible for the enhanced stability against heat and light. However, glycosidation was observed to enhance the stability of anthocyanins against light, but not against heat [51]. The majority of the individual anthocyanins of *S*. *nigra* bear a free hydroxyl group at C-5 and without acyl functionalities in their molecules, while the major anthocyanin individuals of *S*. *canadensis* bear glucosyl residues at C-5 and are acylated by a *p*-coumaroyl functionality. Numerous observations indicate that *S. canadensis*-extracts are more stable to heat and sunlight than *S. nigra* extracts, with the former showing considerable stability to sunlight. The results of a published study indicated that the discrepancy in stability to heat and light between *S. nigra* and *S. canadensis* extracts is ascribed to the structural characteristics of their anthocyanins. The acylation of the anthocyanin molecule by a coumaroyl functionality and/or glycosidation at C-5 led to improved stability of the color of *S. canadensis* extracts. This suggests that glycosidation at C-5 is not an essential factor in heat stability, but that acylation is a crucial factor in heat stability. In addition to acylation, glycosidation at C-5 has some influence on color stability to light. The stability of anthocyanins of wine exposed to light has been reported. The acylated diglucoside anthocyanins were observed to be the most stable, the monoglucosides were the least stable, and the non-acylated diglucosides showed intermediate stability. Additionally, light irradiation of acylated anthocyanins resulted in rapid isomerization from *E* to *Z* form of the *p*-coumaroyl functionality and the subsequent deglucosidation at C-5 of these two isomers. Consequently, it is evident that the acylation of anthocyanins plays a vital role in stabilizing and preserving the original colors of foodstuffs [51].

## 6. Analytical Methods

To maintain consistency and ascertain safety, the elderberry extracts should be subjected to comprehensive characterization and chemical profiling of their phytochemical composition using a range of analytical techniques. Moreover, the stability of the extracts should be determined by analyzing relevant marker constituents in the elderberry extracts. Of the literature published on the analysis of anthocyanins using conventional methods (LC, HPTLC, NMR, IR, and CE), only 2% was published on elderberry [52].

Recently, hyphenated techniques have received attention for addressing a number of analytical problems. Over the years, the power of integrating separation technologies with spectroscopic techniques for both quantitative and qualitative identification of unknown chemicals in complex natural product extracts or fractions has been established. To obtain structural information to identify the compounds present in a crude sample, TLC/HPTLC, LC or HPLC, GC, or CE can be used in combination with spectroscopic detection techniques (e.g., Fourier-transform infrared, photodiode array UV-Vis absorbance, mass spectrometry, or NMR spectroscopy), resulting in the various modern hyphenated techniques (e.g., CE-MS, GC-MS, LC-MS, or LC-NMR).

### 6.1. Infrared Spectroscopy (IR)

Stuppner et al. [53] analyzed TAC in whole elderberries using near-infrared (NIR) spectroscopy. Using quantum chemical calculations, they obtained detailed NIR band assignments of the analyzed compounds and interpreted the wavenumber regions established in partial least squares regression (PLSR) models as meaningful for anthocyanin content. Compared with the pH-differential method and UHPLC-MWD-UHR-Q-TOF-MS for the detection of TAC, the NIR measurement method has proved to be a fast and cost-efficient alternative. It is also a sustainable green technology because it excludes the use of solvents involved in other methods. NIR may prove to be a reliable screening method for the ideal harvest time with maximal content of TAC and the lowest content of cyanogenic glycosides in elderberry.

### 6.2. Nuclear Magnetic Resonance (NMR)

Blunder et al. [54] developed a straightforward and efficient approach using combined NMR and HPLC techniques for the routine analysis of known flavonoids. Proton and multiplicity-edited HSQC spectra were recorded within a significantly shorter time (ca. an order of magnitude) in this study, and the resulting spectra provided enough information for fast structure elucidation, particularly for flavonoid glycosides. This approach can be applied for flavonoid identification in herbal dietary supplements and food quality control.

### 6.3. High-Performance Thin Layer Chromatography (HPTLC)

Krüger et al. [21] investigated the anthocyanin patterns in the dried and fresh fruit of four cultivated and six wild-collected *S. nigra* samples using HPTLC-ESI-MS. The content ranged from 159 to 647 mg/100 g in fresh fruit and from 166 to 2764 mg/100 g in dried fruit for cyanidin 3-*O*-sambubioside, and from 112 to 521 mg/100 g in fresh fruit and 95 to 226 mg/100 g in dried fruit for cyanidin 3-*O*-glucoside. Güzelmeriç et al. [55] analyzed the marker phenolic components (chlorogenic acid, rutin, and isoquercitrin) in dry black elder fruit powder and dietary supplements containing *S. nigra* fruit using HPTLC-PDA and HPLC-DAD. The HPTLC fingerprints showed that most of the marketed products did not contain rutin, chlorogenic acid, or isoquercitrin. Moreover, their fingerprint comparison with the reference was found to be different.

### 6.4. Capillary Electrophoresis

Bridle et al. [56] separated anthocyanins in strawberry and elderberry extracts using HPLC at pH 1.8 and compared these with separations achieved by capillary zone electrophoresis using a standard silica capillary and pH 8.0 running buffer. HPLC separated all the anthocyanins in both extracts, although the similar characteristics of anthocyanins in the elderberry extract made this a more difficult separation due to interference with other compounds. Watanabe et al. [57] reported that micellar electrokinetic chromatography (MEKC) with SDS solutions in a phosphate-borate buffer at pH 7.0 was successful in the separation of elderberry pigments from commercial food samples. Acidic or neutral conditions were employed in this study. HPLC-UV and MEKC-UV gave good separations of these four pigments. However, the HPLC method required longer analysis times using gradient elution, whereas the MEKC analysis was much shorter, not only in separation time, but also in reconditioning the column or capillary.

### 6.5. Gas Chromatography/Mass Spectrometry

GC-MS methods developed for *Sambucus* samples have been used for the analysis of volatile constituents. Using GC-MS, Hale et al. [58] reported 34 volatile compounds yielding 86% of the sample, where aldehydes (phenylacetaldehyde (32.3%) and benzaldehyde (7.9%)) were observed as the main constituents of air-dried mature elderberries. Salvador et al. [22] used GC-MS to analyze the lipophilic components of three Portuguese elderberry cultivars of *S. nigra* (“Bastardeira,” “Sabugueira,” and “Sabugueiro”). The influence of the harvesting season, the cultivar, and the ripening stage was evaluated. The content of the lipophilic extractives ranged from 0.6% to 1.8% (DW). The major compounds were triterpenoids (85–94%) and fatty acids (4–11%). The most abundant compounds identified were ursolic and oleanolic acids, followed by long chain aliphatic alcohols and sterols. Salvador et al. [59] reported on a metabolomic strategy for fingerprinting volatile terpenoids and norisoprenoids from *S. nigra* berries from three cultivars using SPME-GC-ToF-MS. Based on chemometric analysis, 42 monoterpene, 20 sesquiterpene, and 14 norisoprenoid compounds were identified in *S. nigra* berries. Chemometric tools revealed that ripening was the factor that most influenced the volatile fraction profile. Volatile organic compounds (VOC) and essential oils (EO) extracted from different organs of *S. nigra* (leaves, flower buds, flowers, unripe fruit, and ripe fruit) were evaluated by Najar et al. [60]. Jensen et al. [61] identified aroma compounds from samples of seven cultivars of elderberry juice collected using the dynamic headspace technique and analyzed using GC–FID and GC–MS. Vujanović et al. [62] studied the identification and comparison of essential oils obtained from the flowers and fruit of *S. nigra* collected from the Balkan Peninsula. The composition of oils was evaluated using GC-MS, with the oil composition affected primarily by the part of the plant used. The most abundant bioactive compounds in the essential oil of air-dried elderberry fruit were β-damascenone (36%) and linalyl anthranilate (24%). In the essential oil of air-dried elderflowers, the most abundant compound was carane (13%). Vitova et al. [63] identified and quantified the volatile aroma compounds in the fruit of several elderberry cultivars using SPME-GC-MS. Wild-collected elderberries and sixteen cultivars of cultivated elderberries were analyzed. In total, 102 volatile compounds were identified in all elderberry samples, among these 38 alcohols, 16 aldehydes, 10 ketones, 19 esters, four heterocycles, six hydrocarbons, and nine acids. Alcohols, aldehydes, and esters were the most abundant. Significant contents of heterocycles and hydrocarbons were also found. Owing to the highest total content of the selected compounds, the cultivars, “Korsör” (78 mg/kg), “Pregarten” (43 mg/kg), and “Samdal” (68 mg/kg) were used in this study.

### 6.6. Liquid Chromatography/Mass Spectrometry (LC/LC-MS)

For *Sambucus* samples, LC-MS methods are commonly used for the qualitative and quantitative analysis of anthocyanins, phenolic acids, flavonoids, triterpenoids, and cyanogenic glycosides from various parts of the plant material. Various reports have been published, screening the secondary metabolites of *Sambucus* samples and related commercial products, such as juices, jams, dietary supplements, etc. In 1983, Brønnum-Hansen et al. [64] reported on the HPLC-UV/V method for the separation of anthocyanins from *S. nigra* berries under reversed-phase chromatography conditions. *S*. *nigra* and *S. canadensis* were differentiated by the acylated anthocyanin, cyanidin 3-*O*-(6-O-*E*/*Z*-*p*-coumaroyl-2-*O*-β-d-xylopyranosyl)-β-d-glcpyr-5-*O*-β-d-glucopyranoside, which is abundant in the *S*. *canadensis* variety. The separation of acylated anthocyanins was achieved using LC/LC-MS. In 1996, Inami et al. [51] and Lee et al. [37] reported on the differentiation of anthocyanin profiles between *S. nigra* and *S. canadensis* berries, performing stability studies on anthocyanins. Further applications of liquid chromatography methods to separate and study the anthocyanin profiles of commercial elderberry products have been conducted [19,65,66,67] in recent years. These studies help in the quality assessment of juices using elderberries as an active component. So far, various analytical reports have been published on anthocyanin analysis in *S. nigra* [16,37,51,64,68,69,70]. Anthocyanins from elderberries are known for their antioxidant activity. Various reports show the antioxidant activity [19,67,70], anti-microbial activity, enzyme inhibitory activity, anti-glycation, and anti-inflammatory activities of *S. nigra* berries. Table 3 presents a list of liquid chromatography-based analytical methods reported for elderberry fruit.

## 7. Health Effects of European Elderberry and American Elderberry

### 7.1. Link between Antioxidative Activity and Boosting the Immune System

People who are immunocompromised or have a weakened immune system are more vulnerable to infections and other diseases. Low immunity may be ascribed to certain diseases or conditions, such as acquired immunodeficiency syndrome (AIDS), cancer, diabetes, malnutrition, certain genetic disorders, and certain medicines/treatments, such as anticancer drugs, radiation therapy, and stem cell or organ transplants.

An imbalance between reactive oxygen species (ROS)––a result of the immune system response––and the antioxidant defense may lead to the development of oxidative stress-induced systemic inflammation. This condition subsequently impairs immune reactivity, in turn, increasing susceptibility to disease. Polyphenols including anthocyanins possess a powerful antioxidant capacity, and the high intake of dietary anthocyanins or sources rich in anthocyanins, such as elderberry, can mediate the restoration of the balance between oxidants and antioxidants [17,20]. Polyphenols including anthocyanins act by scavenging for the ROS liberated from the immune cells, preventing the self-destruction of immune cells, while also inducing antioxidant enzymes. Antioxidant polyphenols help to restore the reactivity of the immune response, impaired by the excessive generation of ROS [36,74,75].

### 7.2. Effect of Elderberry Fruit Extracts on Oxidative Stress

Free radicals are highly reactive atoms or molecules. ROS and reactive nitrogen species (RNS) are derivatives of oxygen and nitrogen, respectively. ROS comprise O_2_^•−^ superoxide, ^•^OH hydroxyl radical, ^1^O_2_ singlet oxygen, H_2_O_2_ hydrogen peroxide, and ONOO^−^ peroxynitrite [76]. ROS and RNS are generated in all aerobic cells and are implicated in aging as well as in age-related diseases [77]. Despite the damage they cause in cells and tissues, ROS and RNS play an important role in the generation of energy from organic molecules, in immune defense, and in signaling pathways [78]. There are endogenous and exogenous sources of ROS and RNS. The endogenous sources comprise nicotinamide adenine dinucleotide phosphate (NADPH) oxidase (NOX4), myeloperoxidase (MPO), lipoxygenase, and angiotensin II. Superoxide, or radical superoxide anion (O_2_^•−^), is the origin of ROS and RNS, while mitochondria are the major origin of O_2_^•−^ [76]. The predominant source of the radical superoxide anion (O_2_^•−^) is NADPH oxidase. Most of the O_2_^•−^ is dissociated into hydrogen peroxide (H_2_O_2_) by superoxide dismutase (SOD) [78]. H_2_O_2_ is dissociated into the highly reactive hydroxyl ion (^•^OH). Hydroxyl radicals react with phospholipids in cell membranes, lipids, polypeptides, proteins, and nucleic acids.

The exogenous sources of ROS and RNS include oxygen, air, and water pollution, tobacco, alcohol, heavy or transition metals, certain drugs, industrial solvents, and radiation. These substances are metabolized in the body into free radicals [79]. ROS and RNS react with the cellular macromolecules, including carbohydrates, lipids, proteins, and DNA [80]. Antioxidants provide protection against cell and tissue injury by reactive free radicals. This defense system includes endogenous and exogenous antioxidants. Endogenous antioxidants comprise the enzymes, superoxide mutase SOD, catalase (CAT), glutathione peroxidase (GSH-Px), glutathione-S-transferase, and glucose-6-phosphate dehydrogenase [81]. The exogenous non-enzymatic defense system includes ascorbic acid, which is responsible for scavenging for the hydroxyl and superoxide radical anion, α-tocopherol, which protects against lipid peroxidation of cell membranes, vitamin A, glutathione, and polyphenols [76]. Oxidative stress is a natural phenomenon initiated by an imbalance between oxidative agents (ROS and RNS) and antioxidants. This is caused by the excessive production of these oxidative species in the cells and tissues, exceeding the capacity of the antioxidants, or by the diminished ability to scavenge for these radical or to restore the damaged cells or tissues [80]. There is a close relationship between oxidative stress, inflammation, and aging. Chronic oxidative stress and inflammation feed each other and, consequently, increase age-related morbidity and mortality [82,83]. Oxidative stress plays a major role in aging [76,84], and its persistence over a long duration leads to the development of chronic medical conditions, such as diabetes [85], cardiovascular [86,87,88,89] and cardiac disease [90], hypercholesterolemia [91], neurodegenerative disorders [92], inflammation, cancer [93], and autoimmune diseases [94]. Oxidative stress is more likely to happen when the body has low levels of antioxidants. Antioxidants have an enormous effect on preventing, reducing the risk, or attenuating the severity of heart disease, glucose homeostasis, cancer, and other oxidative stress-related disorders [95,96,97,98]. Therefore, the consumption of large quantities of polyphenol-rich vegetables and fruit––rich sources of antioxidant polyphenols––protects against the potential development of oxidative stress-mediated health disorders.

Anthocyanins, which are the major fraction of the constituents of the European elderberry (*S. nigra*) and the American elderberry (*S. canadensis*), are polyphenolic compounds. These are a subclass of flavonoids and are the most commonly occurring flavonoids in food. The proportion of anthocyanins is estimated to be approximately nine times higher than other nutritional flavonoids in certain food products. Anthocyanins are water-soluble glycosides; chemically, they are derivatives of the 2-phenylbenzopyrylium or flavylium salts. Anthocyanins have a distinctive ability to form flavylium cations and acquire different colors, from red to blue or violet, depending on the pH of the medium. The term “anthocyanin” is assigned to the glycoside, while “anthocyanidin” is assigned to the aglycone [99]. Anthocyanins are more stable in acidic solutions (pH 1–3), where they occur as flavylium cations. These colored constituents occur in a variety of berries and fruit [100].

The composition of anthocyanin content (the chemical profile of anthocyanin) is dependent on the cultivar, maturity, season of collection, geographic region, and other factors [101,102]. Anthocyanins have shown the ability to attenuate ROS and RNS in human, animal, and in vitro studies [103,104,105,106]. The therapeutic and health-promoting effects of anthocyanins are attributed to their anti-inflammatory and antioxidant activities. Moreover, anthocyanins contribute to the alleviation of the severity of diabetes, obesity, and cancers via inhibition of the NF-κB-mediated inflammatory pathways. Nevertheless, substantial disparities in the effects of anthocyanins have been observed, primarily due to the structural diversity of anthocyanins [107].

Preclinical studies have demonstrated that extracts of the *S. nigra* berry, which is rich in anthocyanins, exhibit a wide spectrum of bioactivities, including anti-inflammatory, antioxidant, antiviral, antiproliferative, anti-diabetic, and immune-stimulatory activities [48,108], as well as improving blood pressure and dyslipidemia [109]. The neuroprotective effects of elderberry extract were reported in research performed under a neurodegeneration research-driven hypothesis focused on microglial activation or amyloid-β peptide toxicity [108]. In addition, anthocyanins show cytoprotective, antitumor, anti-obesity, and lipidomic effects. Epidemiological evidence indicates the existence of a direct correlation between anthocyanin intake and a lower incidence of chronic and degenerative disorders [110]. The significant antioxidant activity of these polyphenolic constituents, which has been verified at multiple levels of research, constitutes the basis for the underlying mechanism of action in many of their health-enhancing properties. Anthocyanins have been implicated in the regulation of cholesterol metabolism, with the ability to significantly reduce blood levels of cholesterol and triglycerides in the liver [111]. Further, anthocyanins have been reported to exhibit neuroprotective effects, which have been verified based on in vitro and in vivo studies [99,112,113,114].

Moreover, anthocyanins have been reported to play a role in the induction of antioxidant enzymes. In addition to the free-radical scavenging pathway, other mechanisms and pathways are implicated in the pharmacological and health-enhancing effects of anthocyanins, including the cyclooxygenase pathway, the mitogen-activated protein kinase pathway, and inflammatory cytokines signaling [115]. Elderberry fruit extract has reportedly inhibited oxidative stress in a dose-dependent manner, with 1.0 mg/mL of colon-digested elderberry fruit extract exerting an antioxidant activity that was able to protect colon cells against the harmful effects of oxidative stress. This effect was achieved by decreasing the production of excessive intracellular ROS and preventing the oxidative damage of DNA in human mucosal cells of the colon. The major constituents of elderberry fruit extract are the polyphenols, anthocyanins, representing approximately 80% [36]. Oxidative stress is linked to diseases such as diabetes, heart injury, liver injury, and renal dysfunction. ROS produce signaling responses, cause disruption of cellular functions, and lead to tissue damage [116]. Oxidative stress also contributes to the pathogenesis of neurodegenerative diseases [117]. A large body of literature pertaining to in vitro, in vivo, and human trials has demonstrated the health functions of elderberry based on the attenuation of oxidative stress via its antioxidative effect and its anti-inflammatory, anti-diabetic, cardiovascular-protective, and neuroprotective activities. The in vitro and in vivo antioxidant effects of elderberry fruit extract are mediated by regulating antioxidant enzymes (e.g., SOD, GPx, and NOX4), which suppresses intracellular ROS and reduces cell and tissue damage. The pharmacological effects of elderberry fruit extract are ascribed largely to the polyphenolic constituents, anthocyanins, being present as the largest portion of the total constituents of the elderberry fruit [36].

### 7.3. Effect of Elderberry on Huntington’s Disease

Huntington’s disease (HD) is a genetically inherited, progressive brain disorder. It is one of the neurodegenerative diseases. This disease damages the striatum, cerebral cortex, and basal ganglia; however, the striatum is the first area to be affected. HD is manifested by uncontrolled movements, emotional problems, and cognitive decline, later followed by loss of cognitive function. It causes the gradual breakdown and death of nerve cells (neurons) in parts of the brain. Oxidative stress has been demonstrated to play a critical role in the onset of HD, though the exact mechanism of the pathological events has not been elucidated. Elderberry, rich in anthocyanins, which are reported to suppress oxidative stress, was examined in vivo on a rat model as a potential therapeutic candidate for the treatment of HD or the attenuation of the severity of the disease. Thus, in a recently published study on the role of elderberry extract in the treatment of HD, a ground material of lyophilized elderberry fruit was added to the diet of rats injected with 3-nitropropionic acid in an experimental model of HD (the 3-NP-induced rat model of Huntington’s disease) for two months. This was compared with a control group of HD model rats, which had been fed on an ordinary standard diet. Rats fed on the diet containing elderberry showed significant recovery of their motor and muscle coordination in the 3-nitropropionic acid injected rats compared with the control group. In addition, feeding the HD model rats with the elderberry diet resulted in a significant drop in the 3-nitropropionic acid-induced elevated caspase-3 and TNF-α levels. The incorporation of elderberry extract into the diet of the HD model rats significantly enhanced the antioxidative ability of the striatum to suppress ROS generation. There was also a noticeable elevation in the level of glutathione, possibly associated with motor recovery. The underlying mechanism was presumably based on augmenting the antioxidative action and attenuating neuro-inflammation in the HD model rats, properties that anthocyanins are well known to exhibit [118].

### 7.4. Anti-Inflammatory Effects of Elderberry Extract

In an investigation of characterized extracts prepared from the three main Portuguese elderberry cultivars (“Sabugueiro,” “Sabugueira,” “Bastardeira”), the extracts were evaluated in lipopolysaccharide-stimulated RAW 264.7 cells for their potential anti-inflammatory and cellular antioxidant bioactivities. The lipopolysaccharide-stimulated RAW 264.7 cells (monocyte/macrophage-like cells) that were treated with elderberry extract showed dose-dependent inhibition of nitric oxide release, indicating an anti-inflammatory effect of elderberry extract. In another experiment, hepG2 and Caco-2 cells treated with elderberry extract were not affected by the *tert*-butyl hydroperoxide (*t*-BOOH) induced toxicity. In a third experiment on Caco-2 cells that were treated with elderberry extract, it was found that the production of ROS was suppressed and abnormal morphological changes and DNA fragmentation were absent. This experiment provided evidence of the protective role of elderberry extract against the toxicity exerted by *t*-BOOH oxidative stress. The overall results of this research demonstrate the ability of elderberry extract to inhibit and suppress oxidative stress and inflammation [119].

### 7.5. Effects of Elderberry Extract on Diabetes

Dried extract of *S. nigra* fruit was evaluated for its effects on diabetes mellitus using streptozotocin-induced diabetes in rats. The key components of the examined extract were anthocyanins, along with other phenolic constituents. The results of this study showed that the atherogenic index (AI) in diabetic rats was significantly higher than in the healthy groups and higher than the AI for the group of diabetic rats that were administered the elderberry extract. These results demonstrate the ability of elderberry constituents (mainly anthocyanins) to maintain AI values within normal limits. Additionally, the glycosylated hemoglobin in the diabetic rats was reduced due to the administration of the elderberry extract. However, the serum levels of the antioxidants, GSH-Px and GSH, in the diabetic rat group were significantly lower than the levels of the healthy groups and of the group of diabetic rats that were administered elderberry extract. The suppression of endogenous antioxidants is attributed to the oxidative stress developed in response to diabetes. Further, lipid peroxide generation was diminished and LDL oxidation was inhibited by the effects of the elderberry extract. This study demonstrated the antioxidative activity of *S. nigra* fruit extract and its beneficial effects on diabetic rats [120].

### 7.6. Role of Elderberry Anthocyanin in Attenuating Diabetes

An extract of anthocyanins, composed of cyanidin 3-*O*-glucoside, the main component of elderberry extract, was shown to modulate carbohydrate metabolism and glycaemia by inhibiting certain enzymes, such as α-glucosidase and dipeptidyl peptidase 4 (DPP-4). These effects have been demonstrated in vitro, in vivo, and in clinical trials [99,121]. Epidemiological studies associate the consumption of anthocyanins with a lower risk of type 2 diabetes. The anti-diabetic effects of anthocyanins and proanthocyanins were reported in several clinical trials. In addition, anthocyanins can also protect pancreatic cells from glucose-induced oxidative stress [122]. Anthocyanin and anthocyanidin were reported to improve glucose homeostasis by influencing the mass and function of β-cells, insulin sensitivity, and glucose uptake [123].

### 7.7. Elderberry Extract as Adjuvant Therapy for the Treatment of Hypertension

Renin inhibitors are a class of pharmaceutical drugs used to control elevated blood pressure. However, renin inhibitors such as “Aliskiren” cause adverse effects and may cause severe hypotension. A polyphenol extract of *S. nigra* (PP) was evaluated as adjuvant therapy when combined with Aliskiren to augment its efficacy and possibly attenuate its adverse effects. This was accomplished by carrying out an in vivo study on L-NAME-induced hypertension in rats (drug-induced hypertension). The study extended over 8 weeks and investigated the combination effects on blood pressure, lipid profile, and oxidative stress in rats.

This research demonstrated that administration of the combined therapy (the polyphenol extract of *S. nigra* and Aliskiren) resulted in elevating the antioxidant capacity in L-NAME-induced hypertension in rats through the suppression of oxidative stress. Moreover, blood pressure was normalized compared with the severe hypotension showed in the animal group that was fed on a diet devoid of *S. nigra* extract but injected with Aliskiren.

Treatment with the combined therapy showed blood pressure approaching the control group. Additionally, HDL levels in the combined therapy group approached those of the control group and were significantly higher than HDL levels in the induced hypertension group. [109].

### 7.8. Elderberry Extract as Potential Senolytics

The anthocyanins of *S. canadensis* were reportedly shown to have potential antiaging effects since they exhibited antioxidant and anti-inflammatory properties [124]. The primary cause of aging and age-related diseases is cell senescence and, consequently, the accumulation of senescent cells in different tissues. Accelerated generation of ROS results in cellular senescence in addition to cellular apoptosis, necrosis, and autophagy [124,125]. Compelling evidence suggests that aging is the main risk factor in initiating many chronic diseases, disabilities, and declining health. The prevalent age-related diseases include Alzheimer’s disease, Parkinson’s disease, cataracts, macular degeneration, glaucoma, atherosclerosis, hypertension, type 2 diabetes, and cancer. Senescent cells are aged cells that permanently stop division (cell cycle arrest) and show resistance to apoptosis, diminishing the regenerative and reparative capability of tissues. Nevertheless, senescent cells are alive, active, and accumulate in various tissues of the body, releasing harmful substances that inflict inflammation and damage on the surrounding healthy cells. Senescent cells may be implicated in the development of cancers and other age-related diseases [18]. Senescence is activated by several cellular stresses. The formation of senescent cells occurs during the course of life and has been shown to have a beneficial role in different physiological and pathological processes, such as wound healing, tumor suppression, and host immunity. However, the steady production of senescent cells with age has adverse consequences. The senescent cells release pro-inflammatory cytokines, responsible for initiating aging-related diseases and morbidity. Thus, the clearance of senescent cells may ameliorate aging-associated health disorders [18,126].

ROS generation has been implicated in the induction of senescence [127]. Conversely, apoptosis and autophagy pathways have been reported to play pivotal roles in the elimination of senescent cells and damaged tissues [126,127]. The use of senolytics has become a promising approach to delay cellular senescence. Senolytics are drugs that have the ability to selectively induce apoptosis of the senescent cells and reduce age-related degeneration of tissues. The human body needs continuous and controlled cell proliferation to replace senescent cells and damaged cells in order to maintain normal functions. The PI3K/AKT/mTOR pathway is an intracellular signaling pathway that is involved in regulating the cell cycle. Hence, this pathway is implicated in cellular quiescence, proliferation, cancer, and longevity. mTOR is a 289-kDa serine/threonine protein kinase in the PI3K-related protein kinases (PIKK) family. The PI3K/AKT/mTOR signaling pathway has also been reported as a vital pathway in regulating cell senescence [128]. The activation of mTOR signaling initiated cell entry into a phase characterized by increases in cell size and increases in cell number. Additionally, the mTOR pathway regulates the accumulation of biomass and metabolism by modulating essential cellular processes, including protein synthesis and autophagy. Consequently, dysregulation of mTOR signaling has reportedly been involved in metabolic disorders, neurodegeneration, cancer, and aging [128]. Furthermore, significant activation of PI3K/Akt/mTOR signaling has been observed in aging cells, but not in young cells [129]. Thus, it has been suggested that cell senescence could be prevented or reduced via the inactivation of the PI3K/AKT/mTOR signaling pathway [18]. In a recent study, the extract of anthocyanins from *S. canadensis* fruit was assessed for its in vivo and in vitro anti-aging effects using D-galactose-induced senescence in a mouse model. It was found that anthocyanins exhibited the ability to significantly reduce cell senescence and aging of the mouse’s ocular lens. This was achieved by suppressing the activity of PI3K/AKT/mTOR signaling, thereby augmenting the apoptosis of senescent cells, activating autophagic and mitophagic flux, and accelerating the renewal of mitochondria in order to maintain cellular homeostasis. These events attenuated the adverse effects of aging on health. The authors of this investigation suggested the adoption of anthocyanins as new senolytics in age-related health disorders [18].

### 7.9. Role of Elderberry Anthocyanins in Mitigating Mitochondrial Dysfunction

Mitochondria are the source of energy production for the cell in the form of ATP. They play a vital role in the cellular modulation of redox signaling pathways, apoptosis, and autophagic turnover of the constituents of the cell. Mitochondrial dysfunction is one of the causes of cell death. Mitochondrial redox chain dysfunction and oxidative stress are implicated in the onset and progression of several degenerative brain disorders, including Alzheimer‘s and Parkinson’s diseases [130]. In a recent study to evaluate the effect of an elderberry anthocyanin-enriched extract on oxidative stress and mitochondrial dysfunction using rat brain mitochondria and the SH-SY5Y cell line, it was observed that in addition to the powerful antioxidant activity, the anthocyanin-enriched extract displayed an affinity for rat brain mitochondrial membranes. The anthocyanin-enriched extract exhibited the ability to protect the SH-SY5Y cells against the cytotoxicity inflicted by rotenone treatment, demonstrating the neuroprotective potential of anthocyanins. Further, the anthocyanins showed the ability to restore the balance of the cell redox state without affecting the respiratory parameters of brain mitochondria. However, the anthocyanins were able to protect the mitochondrial complex 1 against the rotenone-induced damage, but increased intracellular levels of reactive oxygen species. Additionally, treatment of the SH-SY5Y cells with the anthocyanin-enriched extract elevated the activity of the antioxidant enzymes and mitochondrial respiratory complexes. Consequently, the authors suggested the use of anthocyanin-rich elderberry extract as a potential therapy to mitigate mitochondrial dysfunction, which is an early sign of neurodegeneration. This finding is relevant to the treatment of neurodegenerative diseases, a suggestion that is substantiated by the fact that the antioxidant capacity of elderberry is among the strongest measured in fresh fruits [108].

### 7.10. Role of Elderberry Extract and Cyanidin 3-O-Glucoside in Treatment or Prevention of Vascular Endothelial Dysfunction

Endothelial dysfunction may contribute to the initiation and progression of vascular diseases. Several studies have shown that the administration of antioxidants protects endothelial cells against damage caused by oxidative stress. Oxidative stress and inflammation have been reported to be involved in the dysfunction of the vascular endothelium. Several health disorders, including cardiovascular disease, neurodegenerative disease, and cancer, are attributed, at least in part, to defects in the vascular endothelium. Oxidative stress and inflammation inflict an adverse effect on the vascular endothelium by activating transcription factors such as NF-κB, whose function is dependent on cellular redox status. Studies have examined the incorporation of elderberry extract containing four anthocyanins into the plasma membrane and cytosol of endothelial cells (EC) after incubation for 4 h. These studies assessed the potential antioxidative activity of anthocyanins against several of oxidants. Monoglycoside concentrations were found to be higher than those of diglucosides in both the plasma membrane and the cytosol of the endothelial cells, implying that the uptake of the different anthocyanins is structurally dependent. The elderberry-enriched EC (the EC with incorporated elderberry extract) exhibited significant protective activities against various oxidants, including hydrogen peroxide H_2_O_2_, 2,2′-azobis(2-amidinopropane) dihydrochloride (AAPH), and FeSO_4_/ascorbic acid. Thus, anthocyanins apparently play a role in maintaining EC function and protecting against EC dysfunction inflicted by oxidative stress that may lead to vascular disease [131,132]. Anthocyanins represent approximately 80% of the polyphenols in elderberry. Among these, cyanidin 3-*O*-glucoside is a major component [133], occurring in the ripe fruit of black elderberry at the highest level of concentration (794.13 mg/100 g FW) compared with various other fruit, berries, and vegetables with a range of 0.28–41.52 mg/100 g FW). However, unripe fruit contains 138.72 mg/100 g FW; this is much less than the content of ripe fruit, but still the highest level compared with various other edible sources of anthocyanins [133]. The anthocyanins were reported as potent antioxidants and free radical scavengers, implicated in modulating gene expression and signal transduction pathways.

In one study, cyanidin 3-*O*-glucoside exhibited the ability to protect human endothelial cells against injury induced by TNF-α, a powerful pro-inflammatory agent, and against the adverse effects inflicted by the activation of NF-κB, such as increased gene expression of adhesion molecules, leukocyte adhesion to the endothelium, intracellular accumulation of H_2_O_2_, and the harmful effects of lipid peroxidation products. Thus, this work on the role of cyanidin 3-*O*-glucoside in preventing dysfunction of the vascular endothelium provides additional evidence of the well-documented activity of anthocyanins in counteracting oxidative stress and inflammation. Elderberry extract, with its high content of cyanidin 3-*O*-glucoside and other anthocyanins, can be considered an adjuvant therapy for the prevention or treatment of diseases associated with inflammation and oxidative stress, such as atherosclerosis.

### 7.11. Neuroprotective and Anti-Diabetic Activity of Cyanidin 3-O-Glucoside

In a study to evaluate the neuroprotective and anti-diabetic activity of cyanidin 3-*O*-glucoside, a major anthocyanin in elderberry fruit, in vitro enzyme inhibition bioassays were employed as a method of assessment. The enzymes that are implicated in neuroprotection, namely, monoamine oxidase (MAO-A), tyrosinase TYR, and fatty acid amide hydrolase (FAAH), were employed in these experiments. The enzymes considered as the current targets for type 2 diabetes, namely, α-glucosidase (α-GLU) and dipeptidyl peptidase-4 (DPP-4), were also utilized in this investigation. Cyanidin 3-*O*-glucoside inhibited monoamine oxidase (MAO-A), tyrosinase TYR, and fatty acid amide hydrolase (FAAH) enzymes with IC_50_ of 7.6 μM, 18.1 μM, and 152.1 μM, respectively, while choline esterase was not inhibited. In addition, cyanidin 3-*O*-glucoside exhibited inhibitory activity on α-glucosidase and dipeptidyl peptidase-4 with IC_50_ of 479.8 μM and 125.1 μM, respectively.

The antioxidant activity of cyanidin 3-*O*-glucoside was assessed using the xanthine/xanthine oxidase method. The results showed that cyanidin 3-*O*-glucoside was a more powerful antioxidant than gallic acid, which is considered a standard reference in many antioxidant assays. The results of this study, in addition to previous research, suggest that cyanidin 3-*O*-glucoside should be considered for further investigation in order to assess its therapeutic potential as an antioxidant, neuroprotective, and anti-diabetic [134].

### 7.12. Inhibition of UVB-Induced Oxidative Damage and Inflammation by Cyanidin 3-O-Glucoside

Cyanidin 3-*O*-glucoside, a major anthocyanin in elderberry fruit, has reportedly shown powerful antioxidant and anticarcinogenic properties. The in vivo effects of cyanidin 3-*O*-glucoside on UVB irradiation-induced chronic inflammatory responses were studied in SKH-1 hairless mice. The research results showed that cyanidin 3-*O*-glucoside inhibited UVB-induced skin damage and inflammation in the SKH-1 hairless mice model. Cyanidin 3-*O*-glucoside was able to inhibit the depletion of glutathione and suppress lipid peroxidation and myeloperoxidation in mouse skin caused by chronic exposure to UVB. Further, cyanidin 3-*O*-glucoside significantly diminished the release of UVB-induced pro-inflammatory cytokines, including IL-6 and TNF-α, which are associated with cutaneous inflammation [135].

### 7.13. Anti-Hyperlipidaemic Effect of Elderberry Extract

The level of serum high-density lipoprotein cholesterol (HDL-C) is inversely correlated with the high risk for CVD. High levels of HDL-C protect against the development of atherosclerosis. Atherosclerosis is an inflammatory disease and a responsible factor in the impairment of high-density lipoprotein function, including lowering the levels of HDL-C, HDL antioxidant, and anti-inflammatory activities. The polyphenols, anthocyanins, have been reported to exhibit powerful antioxidant and anti-inflammatory properties. An anthocyanin-rich black elderberry extract was examined for its ability to protect against inflammation-induced HDL dysfunction and against atherosclerosis in apoE−/− mice, a mouse model of hyperlipidemia and HDL dysfunction. The black elderberry extract fed to the mice was rich in cyanidin 3-*O*-sambubioside and cyanidin 3-*O*-glucoside. After 6 weeks, the levels of serum lipid of the anthocyanin-fed mice were not significantly different from the control. Levels of aspartate transaminase and fasting glucose were decreased in the black elderberry extract-fed mice. Changes in the hepatic and intestinal mRNA in the black elderberry extract-fed mice indicated improved HDL function. The levels of hepatic cholesterol were also decreased. Additionally, the activity of serum paraoxonase-1 (PON1) aryl esterase was significantly elevated in the elderberry extract-fed mice. PON1 (paraoxonase 1) is a major anti-atherosclerotic component of HDL. PONI 1 occurs in the circulation in association with HDL, and plays a pivotal role in anti-atherosclerosis thanks to its ability to remove harmful oxidized lipids. In this way, it protects against the development of atherosclerosis. Furthermore, the levels of serum chemokine (C–C motif) ligand 2(CCL2), pro-inflammatory cytokines, were significantly decreased in the elderberry extract-fed mice compared with the control-fed mice. Additionally, in the elderberry-fed mice, the total cholesterol content in the aorta of this group was significantly lowered, indicating reduced atherosclerosis progression. Consequently, this in vivo study demonstrated that elderberry extract may have the potential to attenuate chronic inflammation-mediated HDL dysfunction by influencing hepatic gene expression [136].

### 7.14. Clinical Trials

Numerous clinical trials have been conducted on anthocyanins extracted from berries other than elderberry, and many have shown positive effects on cardiovascular disease, diabetes, and lipid profiles [137,138,139,140,141,142,143]. However, few clinical trials have been carried out on the effects of elderberry, which is rich in anthocyanins, on the abovementioned health disorders. The data from these clinical trials can apparently be extrapolated to elderberry based on the structural similarity of the anthocyanins in all berries, as previously reported [52].

A randomized, double-blind, placebo-controlled trial was conducted to examine the effect of anthocyanin-rich elderberry juice on serum lipid profile. The study was divided into two parts. Part 1 was designed to determine the effect of consumption of anthocyanin-rich elderberry juice on cholesterol and triglyceride blood levels and the resistance of LDL to oxidation. The study was carried out on a cohort of 34 participants. The participants were administered 400 mg capsules containing the powder of spray-dried elderberry juice for two weeks. A subgroup of 14 participants continued taking the capsules of the lyophilized elderberry juice for an additional week to test for resistance of LDL against oxidation. Part 2 was to determine the short-term effect of the anthocyanin-rich elderberry juice on the levels of blood lipids. In this study, each of the six participants was administered a single dose of 50 mL of elderberry juice (equivalent to 10 capsules) together with a high-fat breakfast. The results of this pilot study demonstrated that there was not a significant difference in the blood cholesterol levels between the elderberry group and the placebo group. The resistance of LDL against copper-induced oxidation did not improve with the intake of elderberry juice for three weeks. Additionally, in the single-dose study, the changes in the serum postprandial triglycerides were not significantly different between the elderberry juice plus high-fat breakfast group and the control group. In conclusion, this study indicated that the administration of low doses of lyophilized elderberry juice had a limited effect on lowering serum lipids and did not increase resistance to lipid oxidation [6].

Another randomized, placebo-controlled study was conducted to determine the effect of chronic consumption of anthocyanins (from elderberry) on the biomarkers of CVD risk, liver, and kidney functions. The participants, 52 healthy postmenopausal women (n = 26 in treatment and placebo groups) were administered 500 mg/d anthocyanins as cyanidin glycosides from elderberry or placebo for 12 weeks. The concentrations of anthocyanins and the levels of the biomarkers of CVD (inflammatory biomarkers, platelet reactivity, lipids, and glucose), in addition to the standard parameters of liver and kidney functions, were evaluated in fasted blood at the beginning and at the end of the 12-week study. Anthropometric, blood pressure, and pulse measurements were recorded, and postprandial plasma anthocyanins were measured at different intervals after the administration of a single 500-mg oral dose. The 12-week study demonstrated that chronic intake of 500 mg/d of elderberry extract over 12 weeks is suggested to be safe but did not affect the biomarkers of CVD risk in healthy postmenopausal women [144]. The pharmacological effects and health benefits of elderberry are compiled in Figure 2.

## 8. Safety and Potentially Harmful Compounds in Elderberry

Cyanogenic glycosides (CNGs) are naturally occurring nitrogenous secondary plant metabolites, which consist of an aglycone and a sugar moiety [145,146]. Intact CNGs are nontoxic compounds; however, endogenous plant enzymes react with CNGs releasing hydrogen cyanide (HCN), causing potential toxicity issues [145,146,147,148,149]. Several species of the genus *Sambucus* have previously been shown to be cyanogenic. Among these are *S. glauca* Nutt., *S. racemosa* L., and *S. sieboldiana* Blume exMiquel (seed) [150,151,152]. A species distributed widely in Europe and Middle Asia, *S. nigra* and *S. canadensis* have been shown to contain the cyanogen (*S*)-sambunigrin [153,154], as do the American species, *S. racemosa*, *S. callicarpa,* and *S. microbotrys* [152]. In a subsequent study, both (R)-prunasin and (*S*)-sambunigrin, as well as *m*-hydroxy-substituted glycosides, such as (*S*)-zierin, (*R*)-holocalin, and 6′-acetylholocalin, were reported [155] in *S. nigra*. Materials of *S. caerulea*., *S. ebulis*., *S. gaudichaudiana*, *S. melanocarpa*, and *S. pubens* gave negative tests for cyanide by the picrate method [150]. Buhrmester et al. [154] conducted a study on American elderberry (*S. canadensis*). The presence or absence of cyanogenic glycosides from nine populations from east central Illinois was examined. The study tested for cyanogenic glycosides in the dried leaves. Of the nine elderberry populations examined, only one population tested positive for HCN production for each of the three times tested. In another population, the production of HCN was highly variable. The cyanogenic glycoside was determined to be (*S*)-sambunigrin using gas chromatographic separation of the TMS-derivative. The presence of cyanogenic compounds was quite variable between the populations of European and American elderberry, and these compounds were even absent in many cases [154], while the toxicity of *S. nigra* has been reported to be rare and low [156]. The highest amounts of sambunigrin are present in elder leaves (27.68–209.61 µg/g FW). Lower amounts were found in flowers (1.23–18.88 µg/g FW), whereas unripe berries contain the lowest amounts of this compound (0.08–0.77 µg/g FW). It was also found that the content of sambunigrin in elderberry changes depending on the growing altitude. The highest content of sambunigrin was recorded on hilltops (1048 and 1077 m above sea level), which had lower temperatures and higher solar radiation compared with other altitudes studied (209–858 m above sea level) [157]. In another study by Appenteng et al. [158], the UHPLC-MS/MS and picrate paper methods were used to determine the intact CNGs. No quantifiable trace of cyanide or CNG was detected in commercial elderberry juice. Moreover, traces of CNGs (amygdalin, dhurrin (prunasin + sambunigrin), and linamarin) detected in tissues of American elderberry samples were generally low, with lower levels in the juice and seeds compared with stems and barks.

Other harmful compounds present in elderberry are lectins sharing a high amino acid sequence homology; some of these have *N*-glycosidase activity, which is typical for type II ribosome-inactivating proteins (RIPs). Type II RIPs occur in elderberry bark (e.g., nigrin b-SNA V, SNA I), fruit (nigrin f), and seeds (nigrin s) [159,160]. Moreover, elderberry contains the allergen, Sam n1, which causes type 1 allergy. Some tryptic peptides of Sam n1 demonstrate a high amino acid sequence with lectins and type 2 RIPs located in *Sambucus* [161,162]. However, it was discovered that incubation in a boiling water bath for 5–10 min, made lectins completely sensitive to hydrolytic enzymes in vitro, thus reducing the risk of allergenicity [163]. Lectins and the ribosome-inactivating protein composition of the bark and fruit of elderberry is complex, and the role of these proteins is not well understood [164].

## 9. Stability of Anthocyanins in Elderberry

The fruit of *S. nigra* is an important source of polyphenolic compounds, particularly anthocyanins. Their content in elderberries is relatively high compared with other fruit [165]. The flavonol content of elderberry is much lower than the anthocyanin content. The concentration of quercetin measured in the fruit of thirteen elderberry cultivars ranged from 29 (“Gentofte”) to 60 mg/100 g (“Haschberg”) [166]. Furthermore, *S. nigra* berries contain small amounts of tannins, such as epicatechin (88.4% of total tannins), catechin (11.6% of total tannins), and their thiol derivatives [48].

In Europe, elderflowers and fruit serve as an alternative source in the food industry to produce pies, jellies, jams, ice creams, and yogurts, as well as different beverages, such as wine, tea, liqueur, and juice [167,168]. The predominant anthocyanin in elderberry juice concentrate or must was found to be cyanidin 3-*O*-sambubioside, while cyanidin 3-*O*-glucoside was the most abundant anthocyanin in elderberry wine [167,169] and in the fruit of thirteen elderberry cultivars except for “Allesøe” [166].

The alcoholic fermentation of berries causes changes in the content of phenolic compounds and anthocyanins, which also leads to color changes. The concentration of polyphenols, such as neochlorogenic acid, chlorogenic acid, quercetin 3-*O*-rutinoside, quercetin-3-*O*-glucoside, kaempferol 3-*O*-rutinoside, cyanidin 3-*O*-sambubioside-5-*O*-glucoside, cyanidin 3,5-*O*-diglucoside, cyanidin-3-*O*-glucoside, or cyanidin-3-*O*-rutinoside, was observed to be higher in wine than in must, except for cyanidin 3-*O*-sambubioside, which significantly decreased during alcoholic fermentation. The storage and aging of elderberry wine decreased the content of each analyzed compound, and after three years, the total phenolic content fell by 21%, while the total anthocyanin content was reduced by 94% compared with young wine. Due to the degradation of anthocyanins, the color of elderberry wine also changed from bright-red to brown-red hues. Antioxidant activity was also higher in young wine than in must, and dropped after aging [167]. The method of juice production affects the content of bioactive compounds. For instance, elderberry juice processed using enzymatic treatment (pectinolysis) demonstrated lower average content of most identified phenolic compounds compared with juices produced without enzymatic treatment [170].

The content of anthocyanins in elderberry products is affected by pH, storage time, and temperature. A published investigation reported excessive degradation of the anthocyanins in elderberry fruit due to exposure to steam sterilization of the fruit prior to shipping to the manufacturers for extraction and production of elderberry extracts [171]. Moreover, it has been reported that anthocyanins could not be detected in several of the bulk extracts, apparently due to thermal degradation.

It has been noted that extending storage time and increasing pH and storage temperature decreased the content of anthocyanins in elderberry juice concentrate [172]. In addition to pH, light, oxygen, selected enzymes, and other factors, heat treatment crucially diminishes the level of anthocyanins in plant samples. The treatment duration and temperature significantly influence anthocyanin degradation. Higher stability of anthocyanins has been achieved by using lower temperatures and shorter heating time during processing [173,174].

Elderberry pomace makes up 25–40% of total fruit weight and is very rich in anthocyanins, containing 75–98% of total anthocyanins found in fresh elderberries [175]. The major anthocyanin in elderberry pomace is usually cyanidin 3-*O*-glucoside (14–78 mg/g DW) and the sum of this compound and cyanidin 3-*O*-sambubioside (15–61 mg/g DW) represents approximately 90% of the total anthocyanin content. This ranges from 39 to 153 mg/g of dry matter depending on the extraction method. In contrast, the content of rutin in elderberry pomace is 6–14 mg/g DW [176]. Consequently, elderberry pomace can be utilized to prepare anthocyanin-rich extracts.

## 10. Concluding Remarks

Compelling evidence, including in vitro, in vivo, epidemiological studies, and clinical trials, indicates that the high intake of elderberry is linked to improving human health and preventing or delaying the onset of chronic medical conditions. The health-promoting effects of elderberries are well documented, with elderberry being a very rich source of anthocyanins (approximately 80%).

The most popular species is the European elderberry (*S. nigra*), followed by the American elderberry (*S. canadensis*). The most usable plant part representing the richest source of anthocyanins is the fruit of both elderberry species. The fruit is therefore utilized predominantly for the production of elderberry extracts. The total anthocyanin content of elderberry is dependent on several developmental and environmental factors, resulting in variations in the chemical composition of the different cultivars of elderberry at the qualitative and quantitative levels. The anthocyanins are the bioactive constituents that contribute to the high antioxidant and anti-inflammatory properties of elderberry, and are presumably the underlying mechanism for the observed health benefits of elderberry.

A survey of the literature showed the voluminous preclinical research, including in vitro and animal model studies, that has been conducted to explore and identify the pharmacological effects of elderberry anthocyanins. It is also evident that the pharmacological effects of elderberry, in particular intervening with age-related disorders, neuroprotection, augmenting the immune system, and attenuation of metabolic syndromes such as dyslipidemia, elevated blood sugar, and CVD, are mediated by the ability of anthocyanin to suppress oxidative stress and reduce inflammatory conditions.

Oxidative stress has been associated with the initiation and progression of several health disorders, such as atherosclerosis, type 2 diabetes, neurodegenerative diseases, cognitive impairment, and cancer. The intake of anthocyanin-rich fruit has been demonstrated to confer protection, prevention, or attenuation of the severity of these diseases or to delay their onset. Several molecular mechanisms and pathways are implicated in the observed health-promoting and medicinal effects of anthocyanins. These include the free-radical scavenging pathway; inhibition of the NF-κB-mediated inflammatory pathways; induction of the antioxidant enzymes, including SOD, catalase (CAT), and glutathione peroxidase (GPx); prevention of the oxidative damage of DNA; inhibition of nitric oxide release; inhibition of α-glucosidase and dipeptidyl peptidase 4; inhibition of monoamine oxidase A, tyrosinase, and fatty acid amide hydrolase enzymes; and suppression of the activity of the PI3K/AKT/mTOR signaling pathway and of inflammatory cytokine signaling.

The factors responsible for the variability of the polyphenolic chemical composition of elderberries at the qualitative and quantitative levels include environmental conditions, growing conditions, cultivars, harvesting year, ripening stage of the berry, and cultivation versus wild collection. These factors, when coupled with the effects of processing, lead to inconsistency in the polyphenolic chemical composition of elderberry extracts, which, in turn, results in inconsistency in the pharmacological effects and health benefits of elderberry. The most abundant type of anthocyanidin in the European and the American elderberries is cyanidin. In *S. nigra*, the anthocyanins are not acylated, while in *S. canadensis* the major individual anthocyanins are acylated with *E*-*p*-coumaroyl functionality at C-3. Acylation was reported to stabilize anthocyanins against heat and light. However, glycosidation was observed to improve the stability of anthocyanins against light, but not against heat. Despite the structural variation of the anthocyanins––represented by a number of hydroxyl groups and the substitution of hydroxyl groups with methoxyl groups, as well as the type and number of sugar residues attached to the flavylium cation via glycosidic linkages––the literature reveals that the anthocyanins of different cultivars of elderberry share several pharmacological effects. This observation can be putatively explained by the assumption that the common pharmacological effects of anthocyanins are attributed to the flavylium cation core, which is common for all subclasses of anthocyanins, while the other structural differences, mentioned earlier, contribute to the variation in the magnitude of the effects and the differences in their pharmacokinetic properties. The latter observation may help, in part, to mitigate the inconsistency of the chemical composition of elderberry extracts prepared from different cultivars.

To this end, high-quality clinical trials using elderberry extracts with established chemical profiles are warranted to determine their effects on the plasma biomarkers of CVD risk in a dose–response manner. Moreover, considering the low bioavailability of anthocyanins, additional studies are needed to establish the appropriate doses and the regiment of dosing. Lastly, clinical trials to examine the efficacy of chronic consumption of elderberry extracts are warranted as well.

## Figures and Tables

**Figure 1 molecules-28-03148-f001:**
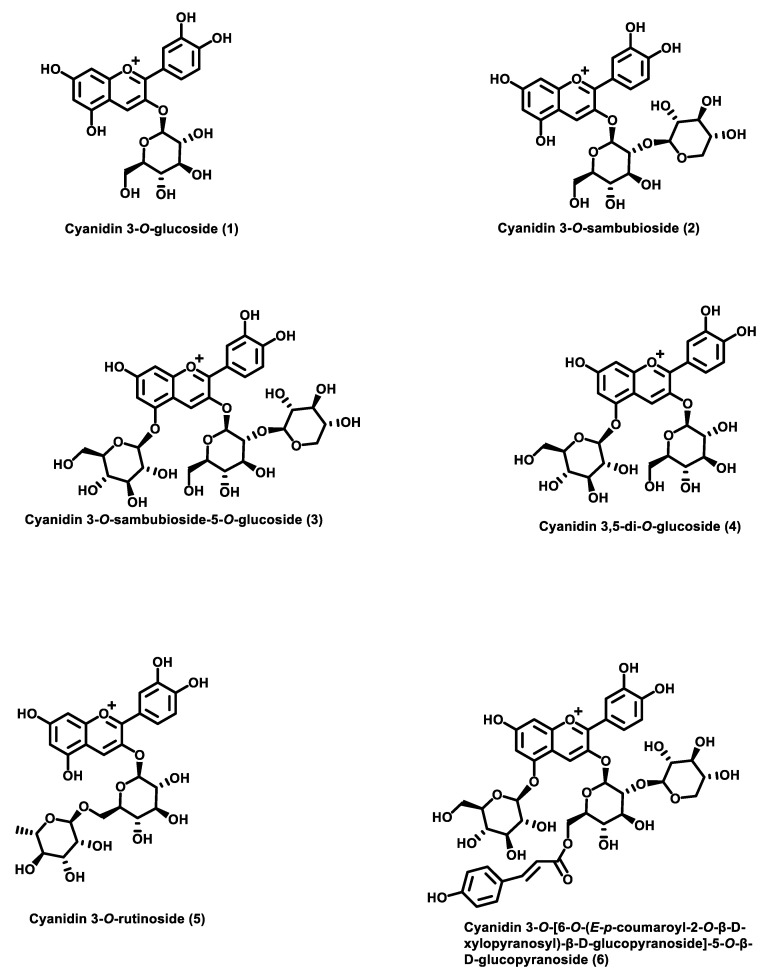
Chemical structure of the anthocyanins of *S. nigra* and *S. canadensis*.

**Figure 2 molecules-28-03148-f002:**
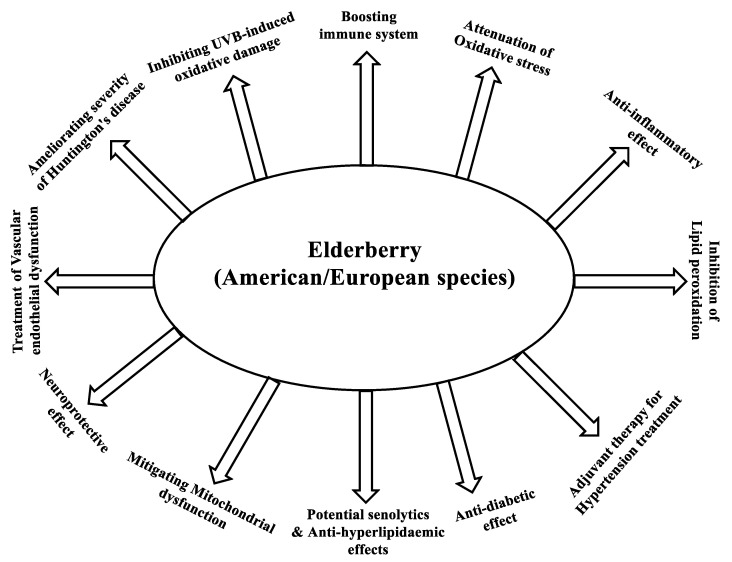
Pharmacological effects and health benefits of elderberry.

**Table 1 molecules-28-03148-t001:** Effect of cultivar and harvesting year (3-year study) on the total polyphenol content and total anthocyanin content.

Type of Effect	Values	Total Polyphenols	Total Anthocyanins
Cultivar	Minimum	852.6 mg/100 g FW	443.6 mg/100 g FW
	Maximum	2541.6 mg/100 g FW	1413.8 mg/100 g FW
Harvesting Year	Minimum	1456.9 mg GAE/100 g	710.5 mg Cy-3-glc/100 g
	Maximum	1740 mg GAE/100 g	895.4 mg Cy-3-glc/100 g

**Table 2 molecules-28-03148-t002:** Effect of harvesting year on the yields of TPC, TAC, and ODPC of the cultivar “Bastardeira”.

Cultivar “Bastardeira”	TPC	ODPC	TAC
Year 1	820 mg GAE/100 g FW	704 mg GAE/100 g FW	510 mg Cy-3-glc/100 g FW
Year 2	1177 mg GAE/100 g FW	703 mg GAE/100 g FW	820 mg Cy-3-glc/100 g FW
Year 3	1016 mg GAE/100 g FW	2009 mg GAE/100 g FW	744 mg Cy-3-glc/100 g FW

**Table 3 molecules-28-03148-t003:** List of analytical methods (HPLC/LC-MS) used to perform qualitative or quantitative analysis of *Sambucus nigra* anthocyanins, along with related commercial products.

No.	Source (Year)	Anthocyanins	Extraction Solvent	% Yield	Conditions		Detection Method	Purpose of Analysis	Pharmacological Activity	[Ref]
					Stationary Phase	Mobile Phase				
1	*S. nigra* berries (1983)	Cy-3-*O*-sam-5-*O*-glc; Cy-3,5-*O*-diglc; Cy-3-*O*-sam; Cy-3-glc;	5 g of sample pomace macerated for 2 min in blender with 100 mL of 0.1 M HCl	-	Nucleosil C18 (150 × 4.6 mm, 5 µm)	A: THF; B: 0.05 M Phosphoric acid; 0–50 min, 1–40% A; 1.2 mL/min	HPLC-UV/Vis	HPLC separation of anthocyanins	-	[64]
2	*S. nigra* and *S. canadensis* berries (1996)	*S. nigra* Cy-3-O-sam-5-O-glc; Cy-3,5-*O*-diglc; Cy-3-*O*-sam; Cy-3-*O*-glc; *S. canadensis* Cy-3-*O*-sam-5-*O*-glc; Cy-3,5-*O*-diglc; Cy-3-*O*-sam; Cy-3-*O*-glc; Cy-3-*O*-(6-O-*E*/*Z*-*p*-cou-2-*O*-β-D-xylopyranosyl)- β -D-glcpyr-5-*O*- β-D-glcpyr; Cy-3-*O*-(6-*O*-*E*-*p*-cou-2-*O*-b-D-xylopyranosyl)-*O*-b-D-glcpyr	500 g of ripe fruit extracted with 2 L of 0.1% HCl in MeOH at room temp. (five repeated extractions)	-	Capcell Pak C18 (250 × 4.6 mm, 5 µm)	A: 0.5% Phosphoric acid in water; B: 0.5% Phosphoric acid in 60% THF 0–30 min, 10–100% B; 1.0 mL/min	HPLC-PDA 520 nm	To study the stability of anthocyanins	-	[51]
3	Elderberry powder (2001)	Cy-3-*O*-sam-5-*O*-glc; Cy-3-*O*-sam; Cy-3-*O*-glc; Cy	Commercial powder containing anthocyanins	-	ODS Hypersil (125 × 4.0 mm, 5 µm)	A: 0.5% Phosphoric acid in water; B: Water/ACN/AA/Phosphoric acid (50:48.5:1:0.5 *v/v*) 0–20 min, 20% B; 20–26 min, 20–60% B; 26–30 min, 60–20% B; 30–35 min, 20% B	HPLC-PDA-MS 520 nm	Anthocyanins method development in botanical supplement raw materials	-	[68]
4	Elderberry juice (2004)	Cy-3,5-*O*-diglc; Cy-3-*O*-sam;	Commercial juice concentrate	-	Capcell Pak C18 UG120 (150 × 4.6 mm, 5 µm)	A: 0.1% TFA in water; B: 0.1% TFA in 50% Acetonitrile 0–60 min, 15–30% B; 0.5 mL/min; 40 °C CT	HPLC-PDA-ESI-MS 520 nm	Chemical profiling of anthocyanins	Antioxidant activity (DPPH)	[19]
5	*S. nigra* and *S. canadensis* berries	Cy-3-*O*-sam-5-*O*-glc; Cy-3,5-*O*-diglc; Cy-3-*O*-sam; Cy-3-*O*-glc; Cy-3-*O*-rut; Pg-3-*O*-glc; Cy based anthocyanin; Dp-3-*O*-rut; Cy-3-*O*-(*Z/E*)-*p*-coumaroyl-sam-5-*O*-glc; Cy-3-*O*-p-coumaroyl-glc; Pt-3-*O*-rut; Cy-3-*O*-*p*-coumaroyl-sam;	5 g of powdered sample extracted using 10 mL of acidified methanol (0.1%FA) for 10 min	Anthocyanins *S. nigra*: 391–806 mg/100 g FW; *S. canadensis*: 207–1005 mg/100 g FW	Synergi Hydro RP (150 × 2 mm, 4 µm)	A: Water: Acetonitrile: AA: TFA—84.8: 5: 10:0.2 *v/v*; B: Acetonitrile; 0–30 min, 99–90% A; 30–40 min, 90–70% A; 40–45 min, 70–60% A; 0.2 mL/min; 25 °C;	LC-MS	Anthocyanins and other polyphenolics variation study	-	[37]
6	*S. nigra* berries (2008)	Anthocyanins, flavonoids, and HCA in berries	80% MeOH extraction solvent	-	Synergi Polar RP (150 × 2 mm)	Fruit: A: Water: Acetonitrile: AA: TFA—50.4:48.5:1:0.1 *v/v*; B: 0.1%TFA in Water; 0–20 min, 20–60% A; 20–21 min, 60–20% A; 21–30 min, 20% A; 30 °C CT;	HPLC-PDA	Phenolic compounds	-	[43]
7	*S. nigra* berries (2013)	Cy-3-*O*-sam-5-*O*-glc; Cy-3,5-*O*-diglc; Cy-3-*O*-sam; Cy-3-*O*-glc;	5 g of berries extracted using acidified methanol (0.3% HCl *v/v*)	Anthocyanins: 272.9 mg/100 g FW	Zorbax SB-C18 (100 × 3.0 mm, 3.5 µm)	A: MeOH; B: 0.1% acetic acid in Water; 0–35 min, 5–42% A; 1.0 mL/min; 48 °C CT;	LC-MS	Chemical characterization of polyphenols	Antioxidant activity (DPPH)	[71]
8	*S. nigra* berries (2014)	Anthocyanins: Cy-3-*O*-sam-5-glc; Cy-3-*O*-sam; Cy-3-*O*-glc;	Fruit extracted with water and left to macerate for an hour	-	Aquasil C18 (150 × 2.1 mm, 5 µm)	A: 0.3% FA in acetonitrile; B: 0.3% FA in Water; 0–50 min, 28% A; 50–60 min, 28–57% A; 60–65 min, 57% A; 0.2 mL/min; 30 °C CT;	LC-MS	Polyphenol patterns in various berry extracts	Antioxidant activity (DPPH)	[72]
9	Elderberry juices (2015)	Cyanidin derivatives; Peonidin derivatives; Pelargonidin derivatives	Commercial juices	Phenolics: 2.2–7.2 mg/mL; Anthocyanins: 0.1–5.3 mg/mL	BEH RP C18 (50 × 2.1 mm, 1.7 µm)	A: 4.5% FA in Water; B: 0.1%FA in acetonitrile; 0–4 min, 5–95%B; 0.4 mL/min;	UPLC-MS/MS	Determination of anthocyanins and total polyphenols	-	[66]
10	*S. nigra* berries (2015)	Cy-3,5-*O*-diglc; Cy-3-*O*-sam-5-glc; Cy-3-*O*-glc; Cy-3-*O*-sam; Cy-3-*O*-rut; Pg-3-*O*-glc; Pg-3-O-sam	70% EtOH–Water solution acidified by acetic acid	Crude extract: 6.0 mg/100 g DW; Purified extract: 48.5 mg/100 g DW	Gemini C18 (150 × 3 mm, 5.0 µm)	A: 0.1% FA in 3% acetonitrile; B: 0.1% FA in 3% Water; 0.25 mL/min; 45 °C CT;	LC-MS/IT-TOF	Identification and quantification of anthocyanins	-	[69]
11	Elderberry juice (2019)	Cy-3-*O*-sam-5-*O*-glc; Cy-3-*O*-sam; Cy-3-*O*-glc	-	Anthocyanin content: 1.1 mg/mL	Aqua C18 (150 × 4.6 mm, 5 µm)	A: 0.1% TFA in Water; B: Acetonitrile; Gradient program;	LC-MSn	Anthocyanin profile of elderberry juice	Antioxidant activity (DPPH)	[67]
12	*S. nigra* fruit (2020)	Cy-hex; Pg-sambu; Cy-pent-hex; Cy-rha-hex; Cy-dihex; Cy-sambu-rha/glc; Dp-dirham-hex	2 g of sample extracted using 10 mL of 1% formic acid in methanol	-	Anthocyanins: Pursuit C18 (150 × 2 mm, 3 µm)	Anthocyanins: A: Water; B: 0.1% FA in MeOH; 0–6 min, 10–15%B; 6–12 min, 15–25%B; 12–16 min, 25–30%B; 16–30 min, 30%B; 30–42 min, 30–100%B; 0.2 mL/min	HPLC-DAD-HRMS	Identification of polyphenolic compounds in berries	-	[73]
13	*Sambucus spp* berries and commercial products and adulterants	Anthocyanins: Cy-3-*O*-sam-5-*O*-glc; Cy-3-*O*-sam; Cy-3-*O*-glc; Cy-3-*O*-(6-*O*-E-*p*-cou-2-*O*-β-D-xylopyranosyl)-β-D-glcpyr-5-*O*-β-D-glcpyr	0.5 g of powdered sample sonicated in 2 mL of acidified methanol (1% FA) for 15 min. Procedure repeated four more times	-	HSS C18 (150 × 2.1 mm, 1.8 µm)	A: 1%FA in Water; B: 1%FA in acetonitrile; 0–8 min, 11–23%B; 8–13 min, 23–35%B; 13–18 min, 35–100%B; 0.135 mL/min; 45 °C CT;	LC-QDa-MS and LC-QToF-MS	Chemical profiling of anthocyanins and flavonoids (qualitative and quantitative)	-	[16]

## Data Availability

The data presented in this study are available on request from the corresponding author.

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
