# Peer review of "Elderberry Extracts: Characterization of the Polyphenolic Chemical Composition, Quality Consistency, Safety, Adulteration, and Attenuation of Oxidative Stress- and Inflammation-Induced Health Disorders"

_molecules, 2023, doi:10.3390/molecules28073148_

Round 1

Reviewer 1 Report

1. The English need improvement since there are some grammatical and syntax errors in the manuscript. For example, in line number 25, the words “to ABC” may be as “to an ABC”; in line number 45, “source” as “a source”; in line number 59, “common” as “the common”; in line number 84, “colorants is” as “colorants are”; in line number 89, “immune” as “the immune”; in line number 99, “used” as “is used”; in line number 106, “diuretic” as “a diuretic”; in line number 113, “stability” as “the stability”; in line number 129, “effect” as “the effect”; in line number 131, “the molecular” as “molecular”; in line number 144, “single-ingredient” as “a single-ingredient”; in line number 157, “consistency” as “the consistency”; in line number 160, “season” as “the season”; in line number 173, “fruits is” as “fruits are”; in line number 124, “year” as “one year”; in line number 179, “lack” as “a lack”; in line number 122, “vast” as “a vast”; in line number 234, “exception for” as “an exception of”; in line number 239, “the second” as “second”; in line number 244, “characteristic for” as “characteristic of”; in line number 228,  “variation” as “a variation”; in line number 292, “berries was” as “berries were”; in line number 297, “chemical” as “the chemical”; in line number 316, “shortage” as “a shortage”; in line number 318, “emergence” as “the emergence”; in line number 335, “themselves has” as “themselves have”; in line number 337, “detection” as “the detection”; in line number 347, “fruits is” as “fruits are”; in line number 428, “anthocyanins is” as “anthocyanins are”; in line number 452, “enhancement” as “the enhancement”; in line number 453, “stability” as “the stability”; in line number 477, “lot” as “a lot”; in line number 480, “fractions has” as “fractions have”; in line number 534, “analysis” as “the analysis”; in line number 459, “more the” as “the”; in line number 574, “HPLC-UV/Vis” as “the HPLC-UV/Vis”; in line number 597, “immune” as “the immune”; in line number 605, “generation” as “the generation”; in line number 616, “development” as “the development”; in line number 614, “powerful” as “the powerful”; in line number 642, “the carbohydrates” as “carbohydrates”; in line number 657, “long” as “a long”; in line number 664, “consumption” as “the consumption”; in line number 673, “distinctive” as “a distinctive”; in line number 685, “in effects” as “in the effects”; in line number 682, “the health” as “health”; in line number 699, “and was” as “and were”; in line number 703, “in induction” as “in the induction”; in line number 706, “cyclooxygenase” as “the cyclooxygenase”; in line number 709, “antioxidant” as “an antioxidant”; in line number 728, “and a brain” as “and brain”; all over the manuscript, “striatum” as “the striatum”; in line number 735, “ability” as “the ability”; in line number 739, “fruits was” as “fruits were”; in line number 745, “significant” as “a significant”; in line number 748, “noticeable” as “a noticeable”; in line number 749, “the motor” as “motor”; in line number 777, “administration” as “the administration”; in line number 794, “of glucose” as “in glucose”; in line number 796, “treatment” as “the treatment”; in line number 801, “employing” as “the employing”; in line number 806, “the L-NAME-induced” as “L-NAME-induced”; in line number 835, “senescent cell ” as “the senescent cells”; in line number 845, “ability for” as “ability to”; in line number 855, “accumulation” as “the accumulation”; in line number 865, “ability” as “the ability”; in line number 869, “attenuation” as “an attenuation”; in line number 893, “the neurodegenerative” as “neurodegenerative”; in line number 901, “administration” as “the administration”; in line number 937, “prevention” as “the prevention”; in line number 961, “irradiation induced” as “irradiation-induced”; in line number 964, “depletion” as “the depletion”; in line number 965,  “that caused” as “caused”; all over the manuscript, “high- density” as “high-density”; in line number, “high” as “a high”; in line number 997, “via influencing the hepatic” as “by influencing hepatic”; in line number 1011, “is by” as “by”; in line number 1015, “population” as “the population”; in line number, “having the N-glycosidase” as “have N-glycosidase”; in line number, “in food” as “in the food”; in line number 1076, “the elderberry” as “elderberry”; “Higher” as “The higher”; in line number 1096, “trials has” as “trials have”; in line number 1096, “high” as “a high”; in line number 1104, “production” as “the production”; in line number 1107, “the quantitative” as “quantitative”; in line number 1108, “the anti-inflammatory” as “anti-inflammatory”; in line number 1143, “stability” as “the stability”; in line number 1144, “number” as “a number”. The grammar mistakes which are not mentioned here are also to be checked and corrected properly.

2. There are some typing mistakes as well, and authors are advised to carefully proof-read the text. For example, in line number 16, the words “Elder berry” may be as “Elderberry”; in line number 16, “health improving” as “health-improving”; in line number 26, “immune enhancing” as “immune-enhancing”; in line number 55, “colds” as “cold”; all over the manuscript, “anti-proliferative” as “antiproliferative”; in line number 88, “elder flowers” as “elderflowers”; in line number 102, “particular the” as “particular, the”; in line number 106, “extracts” as “extract”; in line number 108, “antidepressive” as “antid-epressive”; in line number 113, “extracts” as “extract”; in line number 134, “supressing” as “suppressing”; in line number 151, “includes” as “include”; in line number 155, “elder” as “elders”; in line number 177, “though” as “through”; in line number, “Concering” as “Concerning”; in line number 258, “ripening the” as “ripening, the”; all over the manusript, “half ripe” as “half-ripe”; all over the manusript, “fully ripe” as “fully-ripe”; in line number 268, “half ripening” as “half-ripening”; in line number 323, “relaying” as “relying”; in line number 345, “shows” as “show”; in line number 428, “plants” as “plant”; in line number 436, “characters” as “characteristics”; in line number 537, “air dried” as “air-dried”; in line number 545, “metabolomic based” as “metabolomic-based”; in line number 588, “shows” as “show”; in line number 602, “transplant” as “transplants”; in line number 618, “though” as “through”; in line number 656, “increases” as “increase”; in line number 653, “stress related” as “stress-related”; in line number 705, “health enhancing” as “health-enhancing”; in line number, “” as “”; in line number 682, “health promoting” as “health-promoting”; in line number 716, “causes” as “cause”; in line number 722, “extracts is” as “extracts are”; in line number 734,  “break down” as “breakdown”; in line number 740, “3-NP induced” as “3-NP-induced”; in line number 753, “”extracts as “extract”; in line number 759, “the the anti” as “the anti”; in line number 771, “beside” as “besides”; in line number 787, “anthocyanins, was” as “anthocyanins was”; in line number 797, “antidiabetic” as “anti-diabetic”; in line number 793, “glucose induced” as “glucose-induced”; in line number 797, “Renin inhibitors” as “Renin-inhibitors”; in line number 812, “high density” as “high-density”; in line number 902, “protect” as “protects”; in line number 914, “elderberry enriched” as “elderberry-enriched”; in line number 923, “berries ,” as “berries,”; in line number 931, “H2O2,and” as “H2O2, and”; in line number 936, “an adjuvant” as “adjuvant” ; in line number 990, “oxidised-lipids” as “oxidized lipids”; in line number 996, “attenuating” as “attenuate”; in line number 996,  “the chronic inflammation mediated” as “chronic inflammation-mediated”; in line number 1077, “More over” as “Moreover”; in line number 1083, in line number 1098, “elder berries” as “elderberries”. The typos not mentioned here are also to be checked and corrected properly.

3. Check the abbreviations throughout the manuscript and introduce the abbreviation when the full word appears the first time in the text and then use only the abbreviation (For example, COVID, FDA CFR, nuclear magnetic resonance (NMR), gas chromatography (GC), capillary electrophoresis (CE), etc.,). And it should be in both abstract as well as in the remaining part of the manuscript. Make a word abbreviated in the article that is repeated at least three times in the text, not all words need to be abbreviated.

4. The literature search should be described in detail. The authors are encouraged to include the database, search engines (like PubMed, ScienceDirect, Google scholar etc.,), the keywords used etc., which may be included since it is a review article.

5. The authors may include or address the following for better outcome of the manuscript. How many articles obtained from each of the search engines? What is the inclusion and exclusion criteria? What is the type of article included in this manuscript? How many articles are included for this manuscript? A diagram depicted the literature search should be included for better understanding and outcome (there is no diagram since it is a review).      

6. The full form of the species should be given when the first time appears in both the abstract and in the remaining part of the manuscript and it should be followed by only the first letter of the genus (For example, Sambucus nigra / Sambucus canadensis when the first time appear and followed by S. nigra / S. canadensis).

7. The authors may include if any toxicity studies are available with elderberry for a better understanding of its safety and beneficial effects and concluded the usage of elderberry are safe.

8. The authors are encouraged to include if data available related with clinical trials of elderberry for better outcome of the manuscript.

Author Response

Response to the reviewers’ comments

Reviewer 1

Thank you very much for the valuable and constructive comments. Below is a point-by-point response to the comments raised.

Comment 1:

The English need improvement since there are some grammatical and syntax errors in the manuscript.

Response:

All the grammatical and syntax errors have been corrected according to the reviewer’s changes.

Comment 2:

There are some typing mistakes as well, and authors are advised to carefully proof-read the text.

Response:

All the typing mistakes have been corrected according to the reviewer’s changes. The proof-read of the text was done.

Comment 3:

Check the abbreviations throughout the manuscript and introduce the abbreviation when the full word appears the first time in the text and then use only the abbreviation

Response:

The abbreviations throughout the manuscript were checked in accordance with the reviewer’s instructions.

Comment 4:

  1. The literature search should be described in detail.

Response:

Many parts and a whole section on clinical trials have been added to the manuscript.

Comment 4:

  1. The authors are encouraged to include the database, search engines (like PubMed, ScienceDirect, Google scholar etc.,), the keywords used etc., which may be included since it is a review article.

Response

The search engines which were used for the literature search and for downloading the full articles including SciFinder, PubMed, and Google scholar, in addition to the keywords and the phrases used in the search, have been added at the end of the introduction.

Comment 5:

  1. The authors may include or address the following for better outcome of the manuscript. How many articles obtained from each of the search engines?

Response

Searching the period 2000-2022 using the three search engines resulted in the following: SciFinder led to 785 results for S. nigra, and 43 results for S. Canadensis. Searching PubMed led to 975 results for S. nigra and 17 results for S. Canadensis. Searching Google Scholar led to 24,300 results for  S. nigra, and 13,500 results for S. Canadensis. Of all these results, 80 were selected on the basis of the relevance to the different sections of the review.

 Comment 5:

  1. What is the inclusion and exclusion criteria?

Response

Thank you very much for the reviewer for raising this important point.

The full-text articles were evaluated and screened based on the following inclusion and

exclusion criteria. Inclusion criteria: Papers published in English language. Articles reporting on the fruit/berry as the target plant part since the majority of the elderberry extracts utilized for manufacturing elderberry dietary supplements are made of the berries. Articles reporting on the main two species for manufacturing extracts; Sabmucus nigra (European elderberry) and Sambucus canadensis (American elderberry). Articles reporting only the polyphenolic constituents of the two main species. Articles reporting the pharmacological effects that are based on attenuating the oxidative stress and suppressing the inflammatory conditions.

Exclusion criteria: Species other than Sabmucus nigra and Sambucus canadensis, plant part (plant organ) other than berries (fruits), chemical constituents other than polyphenols, aqueous extracts that contain proteins or polysaccharides. For clinical trials: trials on mixed berries were excludes. Trials combining elderberry with other ingredients such as Zn, vitamin C or Echinacea extracts were excluded. Clinical trials other than those related to the effect of elderberry extracts on the cardiovascular diseases, hyperlipidemia, or diabetes were excluded.

Comment 5

  1. What is the type of article included in this manuscript?

Response

Research papers, review articles, and one book chapter

Comment 5

  1. How many articles are included for this manuscript?

Response

181 Research papers, review articles, and one book chapter

Comment 5

  1. A diagram depicted the literature search should be included for better understanding and outcome (there is no diagram since it is a review).

Response

As this is a single-ingredient review (elderberry), there are no multiple variables to compare between them in a diagrammatic presentation, as we have done in a recently published review on several berries (Molecules, 28 (2), 560, 2023). However, I added the following statement at the end of the analysis section:

Of all the literature on anthocyanin analysis by the conventional methods (LC, HPTLC, NMR, IR, and CE), only 2% was published on elderberry.

Comment 6

The full form of the species should be given when the first time appears in both the abstract and in the remaining part of the manuscript and it should be followed by only the first letter of the genus (For example, Sambucus nigra / Sambucus canadensis when the first time appear and followed by S. nigra / S. canadensis).

Response

All changes have been done accordingly

Comment 7

The authors may include if any toxicity studies are available with elderberry for a better understanding of its safety and beneficial effects and concluded the usage of elderberry are safe.

Response

A section on the safety (7) and another on adulteration (2.8), as a factor of safety are included in the current manuscript.

Comment 8

The authors are encouraged to include if data available related with clinical trials of elderberry for better outcome of the manuscript.

Response

A section (6.14), on clinical trials was added to the current manuscript and the relevant references.

Reviewer 2 Report

The manuscript Elderberry extracts: Characterization of the polyphenolic chemical composition, quality consistency, safety, adulteration, and attenuation of oxidative stress- and inflammation-induced health disorders, by Ahmed G. Osman * , Bharathi Avula , Zulfiqar Ali , Amar G. Chittiboyina , And Ikhlas A. Khan * , Kumar Katragunta  is a comprehensive review about the Sambucus species chemical content of polyphenols and antocyanins, as well as their health benefits and possible clinical uses.

The article is well written and presents interesting facts about the topic.

I do have some observations:

Introduction is very long and has several repeats, I think the readers would benefit from a shorter and more focused introduction.

At section 3- a table with the polyphenolic chemical composition of the Black Elderberry fruits and American Elderberry would be much easier to follow compared with the text.

Section 6.1 is highly general, too long, and does not belong to the structure of the article, I would suggest to shorten it to one phrase that states the role of the balance between ROS/antioxidants in in boosting immunity and prepares the reader for the actual data. There is no remark about the potential negative role of the antioxidants which can become pro-oxidants in a large amount or they can simply inhibit the anti-oxidant mechanisms of the cells.

A diagram of the most important beneficial roles of the Elderberry extracts for human health would help the reader to better understand the content of the review.

Minor observations

Lines 714-715- the species of ROS are more than just H2O2 and O2- , please correct.

English corrections

Line 158 elderberry extract extract…

Line 164-165- an additional preharvest factor it has been found

Line 232- major N

Line 468- was early reported

Lines 654-656- the phrase is unclear

Author Response

Response to the reviewers’ comments

Reviewer 2

Thank you very much for the valuable and constructive comments. Below is a point-by-point response to the comments raised.

Comment 1

Introduction is very long and has several repeats, I think the readers would benefit from a shorter and more focused introduction.

Response

The changes have been made in accordance with the reviewer’s suggestion.

Comment 2

At section 3- a table with the polyphenolic chemical composition of the Black Elderberry fruits and American Elderberry would be much easier to follow compared with the text.

Response

Table 3 shows the polyphenolic composition of the European elderberry (S. nigra) and the American elderberry (S. canadensis). Th purpose of the discussion in section 3-a is to

Show the variation among the different reports at the qualitative and/or the quantitative levels. The variability of the chemical profiles of elderberries, including the European and the American elder fruits are proofs of the causative factors as discussed in section 2, entitled”Factors affecting consistency of the chemical composition of the herbal extracts”.

Comment 3

  1. Section 6.1 is highly general, too long, and does not belong to the structure of the article, I would suggest to shorten it to one phrase that states the role of the balance between ROS/antioxidants in in boosting immunity and prepares the reader for the actual data.

Response

The changes have been made in accordance with the reviewer’s suggestion.

Comment 4

There is no remark about the potential negative role of the antioxidants which can become pro-oxidants in a large amount or they can simply inhibit the anti-oxidant mechanisms of the cells.

Response

I checked the literature and did not find publications reporting elderberry or its anthocyanins as pro-oxidants. However, there are other publications on anthocyanins from berries other than elderberry acting as pro-oxidants. Your suggestion exactly fits a review on anthocyanins from many berries, we recently published in Molecules (Molecules, 28 (2), 560, 2023).  

Comment 5

A diagram of the most important beneficial roles of the Elderberry extracts for human health would help the reader to better understand the content of the review.

Response

The diagram was included at lines 1084-1085 (Fig. 2).

Comment 6

Lines 714-715- the species of ROS are more than just H2O2 and O2- , please correct.

Response

The following ROS are mentioned in section 6.2. (Effects of elderberry fruit extracts on oxidative stress).

O2•− superoxide

HO hydroxyl radical

1O2 singlet oxygen

H2O2 hydrogen peroxide

ONOO peroxynitrite

Comment 7

English corrections

Line 158 elderberry extract extract…

Response

Correction is done

Comment 8

Line 164-165- an additional preharvest factor it has been found

Response

The statement was amended to read:

An additional preharvest factor:  it has been found that the elderberry fruits had higher polyphenolic contents when grown in a well-organized orchard than in the wild.

Round 2

Reviewer 1 Report

1. There are some grammatical, alignment and typographical errors are noted in the manuscript and it should be thoroughly checked and corrected throughout the manuscript. For example,

·         in line number 89, the words “cases the” may be as “cases, the”;

·         all over the manuscript, “anti-proliferative” as “antiproliferative”;

·         in line number 105, “reportedly” as “are reportedly”;

·         in line number 147, “products or” as “product or”;

·         in line number 151, “studying” as “study”;

·         in line number 168 and 169, “on the basis of the relevance” as “based on their relevance”;

·         in line number 203, “in degradation” as “in the degradation”;

·         in line number 213, “plant” as “plant’s”;

·         in line number 352, “and increase” as “and an increase”;

·         in line number 412, “to identification” as “to the identification”;

·         in line number 548, “and lowest” as “and the lowest”;

·         in line number 551, “to routine” as “to the routine”;

·         in line number 572, “the anthocyanins” as “anthocyanins”;

·         in line number 595, “chemometric” as “the chemometric”;

·         in line number 632, “differentiation” as “the differentiation”;

·         in line number 633, “application” as “applications”;

·         in line number, “to development” as “to the development”;

·         in line number 721, “the antioxidant” as “antioxidant”;

·         in line number 774, “lead to” as “leads to”;

·         in line number 795, “for treatment” as “for the treatment”;

·         in line number 851, “lower” as “a lower”;

·         in line number 869, “the oxidative” as “oxidative”;

·         in line number 869, “Also” as “Also,”;

·         in line number 871, “were fed on diet” as “was fed on a diet”;

·         in line number 880, “the combined” as “combined”;

·         in line number 1086, “above referenced” as “above-referenced”;

·         in line number 1094, “spray dried” as “spray-dried”;

·         in line number 1107, “in limited” as “in a limited”;

·         in line number 1134, “causing” as “and causing”;

·         in line number 1174, “elderberries is” as “elderberries are”;

·         in line number 1270, “inconsistency” as “the inconsistency”.

2. In line number 62, the expansion for COVID-19 is not given and it should be checked properly for all other abbreviations used in the manuscript. For example, in line numbers 1110 and 1111, the authors have used both the abbreviation and expansion for cardiovascular disease (CVD), but in the remaining part of the manuscript, the authors have used only expansion “cardiovascular disease”.

Author Response

Response to the reviewers’ comments, second round

Reviewer 1

Thank you very much for your effort and additional comments. Below is a point-by-point response to the comments raised.

Comment 1

There are some grammatical, alignment and typographical errors are noted in the manuscript and it should be thoroughly checked and corrected throughout the manuscript.

Response

All the grammatical, alignment and typographical errors have been corrected, with three exception as follows:

  1. In line number 213 (now 187), “plant” as “plant’s” . The original phrase: “plant raw material”, is correct.
  2. In line number 774 (now 700), “lead to” as “leads to”. The original statement: ROS produce signaling responses, cause disruption of cellular functions, and lead to tissue damage”, is correct, since the subject is plural (ROS).
  3. In line 1174 (now 1068), “elderberries is” as “elderberries are”. The original statement: Lectins and ribosome-inactivating protein composition of the bark and fruits of elderberries is complex”, is correct since the subject (composition) is singular.

Comment 2

In line number 62, the expansion for COVID-19 is not given and it should be checked properly for all other abbreviations used in the manuscript. For example, in line numbers 1110 and 1111, the authors have used both the abbreviation and expansion for cardiovascular disease (CVD), but in the remaining part of the manuscript, the authors have used only expansion “cardiovascular disease”.

Response:

  1. The acronym Covid-19 stands for coronavirus disease of 2019, and the full name has been inserted before the acronym, in line 62 (now line 61).

  1. in lines 951, 1008, and 1142 (current copy), the full name of cardiovascular diseases was removed and replaced with the abbreviation CVD, and also all through the whole manuscript, except for the first mention.

The linguistic errors have been corrected by consulting an English language professional.

Reviewer 2 Report

The authors have made the changes, in my opinion, the article can be published in the present form. 

Author Response

Reviewer 2

Comments and Suggestions for Authors

The authors have made the changes, in my opinion, the article can be published in the present form. 
